# Diversify & Conquer: Outcome-directed Curriculum RL via Out-of-Distribution Disagreement

**Daesol Cho, Seungjae Lee, and H. Jin Kim**
Seoul National University
Automation and Systems Research Institute (ASRI)
Artificial Intelligence Institute of Seoul National University (AIIS)
dscho1234@snu.ac.kr, ysz0301@snu.ac.kr, hjinkim@snu.ac.kr

## Abstract

Reinforcement learning (RL) often faces the challenges of uninformed search problems where the agent should explore without access to the domain knowledge such as characteristics of the environment or external rewards. To tackle these challenges, this work proposes a new approach for curriculum RL called **D**iversify for **D**isagreement & **C**onquer (**D2C**). Unlike previous curriculum learning methods, D2C requires only a few examples of desired outcomes and works in any environment, regardless of its geometry or the distribution of the desired outcome examples. The proposed method performs diversification of the goal-conditional classifiers to identify similarities between visited and desired outcome states and ensures that the classifiers disagree on states from out-of-distribution, which enables quantifying the unexplored region and designing an arbitrary goal-conditioned intrinsic reward signal in a simple and intuitive way. The proposed method then employs bipartite matching to define a curriculum learning objective that produces a sequence of well-adjusted intermediate goals, which enable the agent to automatically explore and conquer the unexplored region. We present experimental results demonstrating that D2C outperforms prior curriculum RL methods in both quantitative and qualitative aspects, even with the arbitrarily distributed desired outcome examples.

## 1 Introduction

Reinforcement learning (RL) has great potential for the automated learning of behaviors, but the process of learning individual useful behavior can be time-consuming due to the significant amount of experience required for the agent. Furthermore, in its general usage, RL often involves solving a challenging uninformed search problem, where informative rewards or desired behaviors are rarely observed. While there are some techniques that can alleviate the exploration burden such as reward-shaping [30] or preference-based reward [21, 6], they often require significant domain knowledge or human intervention. This makes it difficult to utilize RL directly for problems that require challenging exploration, especially in many domains of practical significance. Thus, it is becoming increasingly crucial to tackle these challenges from the algorithmic level by developing RL agents that can learn autonomously with minimal supervision.

One potential approach is a curriculum learning algorithm. It involves proposing a carefully designed sequence of curriculum tasks or goals for the agent to accomplish, with each step building upon the previous one to gradually progress the curriculum. By doing so, it enables the agent to automatically explore the environment and improve the capability in a structured way. Previous studies primarily involve a mechanism to adjust the curriculum distribution by maximizing its entropy within the explored region [35], or taking into account the difficulty level [43, 11], or learning progress of the agent [36]. However, efficient desired outcome-directed exploration is not available under these

37th Conference on Neural Information Processing Systems (NeurIPS 2023).

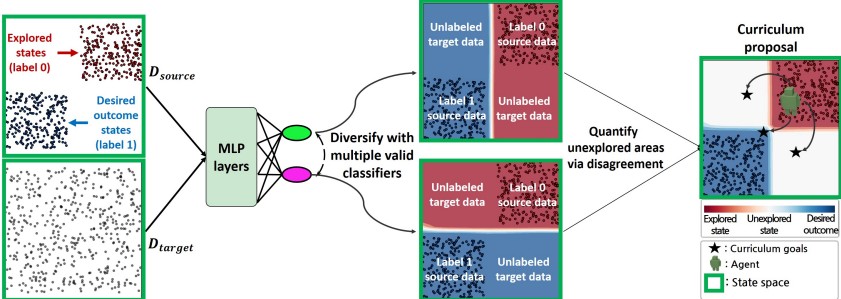

Figure 1: D2C trains a set of classifiers with labeled source data while diversifying their outputs on unlabeled target data (red: predicted label 0, blue: predicted label 1). Then, it proposes curriculum goals based on the diversified classifier's disagreement and similarity-to-desired outcome.

frameworks as these approaches do not have converging curriculum objectives, resulting in a naive search for unseen states.

Providing the agent with examples of successful outcomes can make the RL problem more manageable, as opposed to letting the agent explore without a particular objective and with an undirected reward. Such examples can offer considerable guidance on how to achieve a task if the agent can estimate the similarities between the desired outcome examples and visited states. In the realm of desired outcome-directed RL, previous studies try to maximize the probability of reaching desired outcomes [12, 41, 8]. However, these approaches do not have tools for explicitly quantifying an unexplored region, leading the agent to settle for reaching an initially discovered desired outcome example, rather than continuing to explore further to find another example. Other works try to minimize the distance between the generated curriculum distribution and the desired outcome distribution to propose intermediate task goals [37, 18]. But, these approaches have mainly been confined to problems that do not involve significant exploration challenges because they rely on the assumption that the Euclidean distance metric can represent the geodesic interpolation between distributions. This assumption is not universally applicable to all environmental geometries, making these methods less versatile than desired.

Therefore, it is necessary to develop an algorithm that enables the agent to automatically perform the outcome-directed exploration by generating a sequence of curriculum goals, which can be applied to arbitrary geometry and distribution of the given desired outcome examples. To do so, we propose **D**iversify for **D**isagreement & **C**onquer (**D2C**), which only requires desired outcome examples and does not require prior domain knowledge such as 1) the geometry of the environment 2) or the number of modes or distribution of the desired outcomes, or 3) external reward from the environment. Specifically, D2C involves *diversifying* the goal-conditioned classifiers to identify the similarities between the visited states and the desired outcome states. This is accomplished by ensuring that the outputs of each classifier *disagree* on unseen states, which not only allows determining the unexplored frontier region but also provides an arbitrary goal-conditioned intrinsic reward. Based on such conditional classifiers, we propose to employ bipartite matching to define a straightforward and easy-to-comprehend curriculum learning objective, which produces a range of well-adjusted curriculum goals that interpolate between the initial state distribution and arbitrarily distributed desired outcome states for enabling the agent to *conquer* the unexplored region.

To sum up, our work makes the following key contributions.

- We propose a new outcome-directed curriculum RL method that only needs a few arbitrarily distributed desired outcome examples and eliminates the need for external reward.

- To the best of our knowledge, D2C is the first algorithm for curriculum RL that allows for automatic progress in any environment without being restricted by its geometry or the desired outcome distribution by proposing diversified conditional classifiers.

- In various goal-conditioned RL experiments, our method consistently outperforms the previous curriculum RL methods through precisely calibrated guidance toward the desired outcome states in both quantitative and qualitative aspects.

## 2   Related Works

Despite various attempts to improve exploration in RL, it remains a difficult issue that has not been fully solved. Some previous works for exploration have suggested methods based on information theoretical approaches [9, 39, 50, 20, 17, 27], maximizing the state visitation distribution's entropy [47, 25, 26], counting the state visitation [3, 31], utilizing similarity or curiosity [33, 45], and quantifying uncertainty through prediction models [4, 34]. Other approaches provide a curriculum to allow the agent to explore the environment through intermediate tasks. Curricula are usually created by adjusting the distribution of goals to cover new and unexplored areas. It is achieved by considering an auxiliary objective such as entropy [35] or disagreement between the model ensembles [49, 15, 28] or difficulty level of the curriculum [11, 43], regret [16], and learning progress [36]. However, these methods only focus on visiting diverse frontier states or do not provide a mechanism to converge toward the desired outcome distribution. In contrast, our method enables more efficient outcome-directed exploration via curriculum proposal with an objective to converge rather than simply exploring various frontier states, only requiring a few desired outcome examples.

Assuming access to desired outcome samples or distribution, some prior methods try to accomplish the desired outcome states by maximizing the probability of reaching these states [12, 41, 8]. However, they lack a mechanism for quantifying an under-explored region and synthesizing the knowledge acquired from the agent's experiences into versatile policies that can accomplish novel test goals. Some algorithms generate curricula as an interpolation between the distribution of desired target tasks and auxiliary tasks [37, 18], but they still rely on the Euclidean distance metric, which is insufficient to handle arbitrary geometric structures. There exists a work that addresses geometry-agnostic curriculum generation using the given desired outcome states, similar to our approach [5]. But, it requires carefully tuned Wasserstein distance estimation that depends on the optimality of the agent. It leads to inconsistent estimation before the convergence, resulting in numerically unstable training, while our method does not have such dependence. Also, it adopts a meta-learning-based technique [10, 23] that requires gradient computation at every optimization iteration, while our method only requires a single neural network inference, leading to much faster curriculum optimization.

A core idea behind our method is utilizing classifiers to learn a diverse set of hypotheses that minimize the loss on source inputs but make differing predictions on target inputs [32, 22]. It is related to ensemble methods [7, 19, 14] that aggregate the multiple functions' predictions, but the proposed method is distinct in terms of directly optimizing on an underspecified target dataset for enhancing diversity. Even though this diversifying strategy for target data is typically considered from the perspective of out-of-distribution robustness in conditions of distribution shift [29, 24, 38] or domain adaptation [44, 46, 42], we demonstrate how it can be applied for classifiers to quantify the similarity between the visited states and desired outcome states, and discuss its links to exploration and curriculum generation in RL.

## 3   Preliminary

We consider the Markov decision process (MDP) $\mathcal{M} = (\mathcal{S}, \mathcal{G}, \mathcal{A}, \mathcal{P}, \gamma)$, where $\mathcal{S}$ indicates the state space, $\mathcal{G}$ the goal space, $\mathcal{A}$ the action space, $\mathcal{P}(s'|s, a)$ the transition dynamics, and $\gamma$ the discount factor. In our framework, the MDP is not provided a reward function and we consider a setting where only the desired outcome examples $\{g_k^+\}_{k=1}^K$ from the desired outcome distribution $p^+(g)$ are given. Thus, our method utilizes an intrinsic reward $r : \mathcal{S} \times \mathcal{G} \times \mathcal{A} \to \mathbb{R}$. Also, we represent the curriculum distribution obtained by our method as $p^c(s)$.

### 3.1   Acquiring knowledge from underspecified data

For diversification-based curriculum RL, we train a classification model $y = f(x)$ in a supervised learning setting where $x \in \mathcal{X}$ is input, and $y \in \mathcal{Y}$ is the corresponding label. The model $f$ is trained with a labeled source dataset $\mathcal{D}_S \sim p_S(x, y)$ that includes both the given desired outcome examples ($y = 1$) and the visited states in the replay buffer $\mathcal{B}$ of the RL agent ($y = 0$). We assume that the desired outcome distribution can be modeled as a mixture of outcome distribution, $p^+(x) = \sum_{o \in \mathbb{O}} w_o p_o(x)$, where each $o \in \mathbb{O}$ corresponds to a specific outcome distribution $p_o(x)$. The selection of model $f$ from hypothesis class $f \in \mathcal{F}$ is achieved by minimizing the predictive risk $\mathbb{E}_{p_S(x,y)}[\mathcal{L}(f(x), y)]$, where $\mathcal{L}$ is a standard classification loss.

Although $f$ demonstrates good generalization on previously unseen data acquired from the source distribution $p_S(x, y)$, it is ambiguous to evaluate the model $f$ in distribution shift conditions such as when querying target data obtained from an out-of-distribution (e.g. a state lies in an unexplored region). This is because there could be many potential models $f$ that can minimize the predictive risk. To formalize this intuition, we introduce the concept of an $\varepsilon$-optimal set defined as $\mathcal{F}^\varepsilon :=$ $\{f \in \mathcal{F} \mid \mathcal{L}_p(f) \leq \varepsilon\}$ [22], where $\mathcal{L}_p$ represents the risk associated with a distribution $p$, and $\varepsilon \geq 0$.

The definition of the $\varepsilon$-optimal set implies that the predictions of any two models $f_1, f_2$ in the set are almost identical on $p_S(x, y)$ when $\varepsilon$ is small. But, using $\mathcal{F}^\varepsilon$ as it is has a few drawbacks: 1) there is no criterion to prefer any particular hypotheses of $\mathcal{F}^\varepsilon$ over another, and 2) the trained model $f$ might not be suitable for evaluating the target data distribution as it does not have a mechanism to quantitatively discriminate unseen target data from the labeled source data.

To address this point, we utilize an unlabeled target dataset $\mathcal{D}_T \sim p_T(x)$ for comparing the functions within $\mathcal{F}^\varepsilon$ by examining how their predictions differ on $\mathcal{D}_T$. Since $\mathcal{D}_T$ can be viewed as indicating the directions of functional change that are most crucial to the automatic exploration of the RL agent, we set $p_T(x)$ as the uniform distribution between the lower and upper bound of the state space to include all possible candidate states to visit. Knowing the state space's bounds is a commonly utilized assumption in many curriculum learning works [37, 18, 5] and it does not require being aware of the dynamically feasible areas. Then, our objective is to identify a set of classification models that perform well in $p_S(x, y)$ and disagree in $p_T(x)$ to recognize the unseen state from the unexplored region by quantifying the similarity between the observed states in $\mathcal{B}$ and desired outcome examples. Such a function will lie within $\mathcal{F}^\varepsilon$ of $p_S(x, y)$, and the model leverages $\mathcal{D}_T$ to identify a diverse set of functions within the near-optimal set.

## 4 Method

For an automatic exploration toward the desired outcome distribution via calibrated guidance of the curriculum, the proposed **D2C** suggests curriculum goals via diversified classifiers that disagree in the unexplored region and enables the agent to conquer this area by exploring through the goal-conditioned shaped intrinsic reward. It allows the agent to make progress without any prior domain knowledge of the environment such as obstacles or distribution of the desired outcome states and advances the curriculum towards the desired outcome distribution $p^+(g)$.

### 4.1 Diversification for disagreement on underspecified data

For the automatic exploration of the RL agent with the curriculum proposal, quantification of whether a queried state is already explored or not is required. To obtain such a quantification, we diversify a collection of classifier functions by comparing predictions for the target dataset while minimizing the training error as briefly described in Section 3.1. The intuition behind this is that diversifying predictions will produce functions that disagree with data in the ambiguous region [22].

Specifically, we use a multi-headed neural network with $N$ heads to train diverse multiple functions. Each head $i$ produces a prediction for an input $x$ represented as $f_i(x)$. To ensure that every head has a low predictive risk on $p_S(x, y)$, we minimize the cross-entropy loss for each head using $\mathcal{L}_{\text{xent}}(f_i) = \mathbb{E}_{x,y \sim \mathcal{D}_S}[\mathcal{CE}(f_i(x), y)]$. Ideally, it is desirable for each function to rely on distinctive predictive features of the input to encourage differing predictions. Therefore, we train each pair of heads to generate statistically independent predictions, which implies disagreement in predictions. It could be achieved by minimizing the mutual information between each pair of heads:

$$\mathcal{L}_{\text{MI}}(f_i, f_j) = \mathbb{E}_{x \sim \mathcal{D}_T}[D_{\text{KL}}(p(f_i(x), f_j(x)) \| p(f_i(x)) \otimes p(f_j(x)))] \tag{1}$$

where the input data is obtained from target dataset $\mathcal{D}_T$. For implementation, we compute empirical estimates of the joint distribution $p(f_i(x), f_j(x))$ and the product of the marginal distributions $p(f_i(x)) \otimes p(f_j(x))$, which can be computed using libraries developed for deep learning [22].

In summary, the overall objective for classifier diversification is represented with a hyperparameter $\lambda$:

$$\sum_i \mathcal{L}_{\text{xent}}(f_i) + \lambda \sum_{i \neq j} \mathcal{L}_{\text{MI}}(f_i, f_j) \tag{2}$$

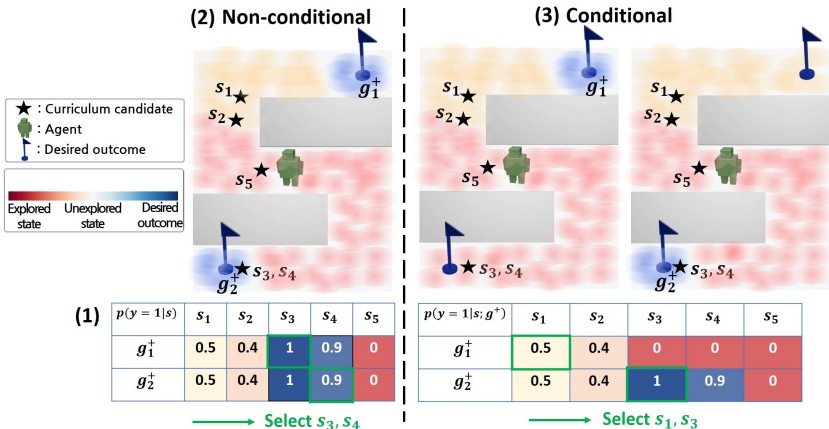

Figure 2: Overview of the curriculum proposal when we select two candidates for curriculum goals of $g_i^+$ by bipartite matching. (1) The curriculum goals are proposed according to the quantification of the similarity-to-desired outcome ($s_i$ with a high probability will be selected). (2) If the classifier is not in the conditional form, it cannot distinguish the source of the desired outcome example $g_i^+$, resulting in collapsed curriculum proposals, (3) while the conditional classifier enables non-collapsed curriculum proposals even when the agent achieves a specific desired outcome (e.g. $g_2^+$) first.

### 4.2 Quantifying unexplored regions by conditional classifiers

From this section, we slightly abuse the notation $s$ instead of $x$ for the RL setting. Considering the diversification process in Section 4.1, we can quantify how much a queried state $s$ is similar to the desired outcome example from $p^+(g)$. Specifically, we can define a pseudo probability of the queried point $s$ by averaging predictions of each head:

$$p_{\text{pseudo}}(y = 1|s) := \frac{1}{N} \sum_{i=1}^{N} f_i(s) \tag{3}$$

When the queried state $s$ is in proximity to the data within $p_S$ (with the desired outcome example labeled as 1 and states from the replay buffer $\mathcal{B}$ labeled as 0), it becomes challenging for the classifiers to give a high probability to labels that substantially differ from the neighboring data, since each classifier $f_i$ is trained to minimize the loss for the source data. On the other hand, if the queried state $s$ is significantly different from the data in $p_S$, each prediction head will output different values since the classifiers are trained to diversify the predictions on the target data (uniform distribution on the state space) to make their prediction values disagree as much as possible. For example, in the case of two heads, one predicts 1 and the other predicts 0 for the same queried target data, resulting in the pseudo probability of 0.5. This could be interpreted as a quantification of how much disagreement exists between classifiers, or uncertainty of the queried data, which can be utilized as an unexplored region-aware classification (Figure 1).

Thus, we can consider a curriculum learning objective for interpolating the curriculum distribution from the initial state distribution to $p^+(g)$ represented by the following cross-entropy loss:

$$\mathcal{L}_{curr} = \mathbb{E}_{s \sim \mathcal{B}, g^+ \sim p^+(g)} \left[ \mathcal{CE}(p_{\text{pseudo}}(y = 1|s); y = p_{\text{pseudo}}(y = 1|g^+)) \right] \tag{4}$$

Intuitively, before discovering the desired outcome examples in $p^+(g)$, the curriculum goal candidate $s \sim \mathcal{B}$ that minimizes Eq (4) is proposed in the frontier of the explored regions where classifiers $f_i$ disagree. And, as the agent explores and discovers the desired outcome examples, the curriculum candidate is updated to converge to $p^+(g)$ to minimize the discrepancy between the predicted labels of $g^+$ and $s$.

However, it is not applicable for a case when the desired outcome examples are spread over multi-modal distribution or arbitrarily because the trained classifiers $f_i$ do not distinguish which $p_o$ the desired outcome example is obtained from (Figure 2). In other words, $f_i$ will predict a value close to

1 for any desired outcome example $g^+$, and the loss in Eq (4) will be close to 0. As the curriculum goal candidates are obtained from the replay buffer $\mathcal{B}$, the curriculum optimization may collapse if the agent achieves one of the desired outcome distributions ($p_o$) earlier, which is not desirable for achieving all the desired outcome examples regardless of its distribution.

Since we assume that we do not know which $p_o$ the desired outcome example is obtained from, nor the number of modes ($o$) of the desired outcome distributions, we propose to utilize a conditional classifier to address this point while satisfying the assumption. Specifically, we define goal-conditioned classifiers, where each classifier takes input $s$ and is conditioned on $g$, and these classifiers are trained to minimize the following modified objective of Eq (2):

$$
\mathbb{E}_{g \sim \mathcal{D}_{\mathrm{G}}} \left[ \mathbb{E}_{s \sim \mathcal{B}} \left[ \sum_i \mathcal{L}_{\mathrm{xent}} \left( f_i(s; g), y = 0 \right) \right] + \mathbb{E}_{\varepsilon} \left[ \sum_i \mathcal{L}_{\mathrm{xent}} \left( f_i(g + \varepsilon; g), y = 1 \right) \right] \right.
$$
$$
\left. + \lambda \mathbb{E}_{s \sim \mathcal{D}_{\mathcal{T}}} \left[ \sum_{i \neq j} \mathcal{L}_{\mathrm{MI}} \left( f_i(s; g), f_j(s; g) \right) \right] \right]
\tag{5}
$$

where $\varepsilon$ is small noise (from uniform distribution around zero or standard normal distribution with a small variance) for numerical stability, and $\mathcal{D}_{\mathrm{G}}$ can be either $\mathcal{D}_{\mathrm{T}}$ or $\mathcal{B} \cup p^+(g)$ for training arbitrary goal-conditioned & unexplored region-aware classifiers. We found that there is no significant difference between these choices. Then, the pseudo probability can be represented as $p_{\mathrm{pseudo}}(y = 1|s; g) := \frac{1}{N} \sum_{i=1}^{N} f_i(s; g)$ and we can address the arbitrarily distributed desired outcome examples without curriculum collapse (Figure 2). This proposed conditional classifier-based quantification is one of the key differences from the previous similar outcome-directed RL methods [23, 5]. Because the previous works require computing gradients of thousands of data for meta-learning-based network inference, while our method only requires a single feedforward inference without backpropagation which leads to fast computation.

### 4.3 Curriculum optimization via bipartite matching

As we assume that we have access to desired outcome examples from $p^+(g)$ instead of their explicit distribution, we can approximate it using the sampled set $\hat{p}^+(g)$ ($|\hat{p}^+(g)| = K$). Then, the problem is formulated by the combinatorial setting that requires finding the curriculum goal candidate set $\hat{p}^c(s)$ that will be assigned to each sample of $\hat{p}^+(g)$, and it can be solved via bipartite matching. With curriculum goal candidates and desired outcome examples, the curriculum learning objective is represented as follows:

$$
\min_{\hat{p}^c(s):|\hat{p}^c(s)|=K} \sum_{s_i \in \hat{p}^c(s), g_i^+ \in \hat{p}^+(g)} w(s_i, g_i^+)
\tag{6}
$$
$$
w(s_i, g_i^+) := \mathcal{CE}(p_{\mathrm{pseudo}}(y = 1|s_i; g_i^+); y = p_{\mathrm{pseudo}}(y = 1|g_i^+; g_i^+))
\tag{7}
$$

The intuition behind this objective is similar to Eq (4), but it is different in terms of considering conditional quantification, which enables addressing arbitrarily distributed desired outcome examples. Then, we can create a bipartite graph $\mathbf{G}$ with edge costs $w$ by considering the sets of nodes $\mathbf{V}_a$ and $\mathbf{V}_b$, which represent achieved states in replay buffer $\mathcal{B}$ and $\hat{p}^+(g)$, respectively. We define the bipartite graph $\mathbf{G}(\{\mathbf{V}_a, \mathbf{V}_b\}, \mathbf{E})$ with edge weights $\mathbf{E}(\cdot, \cdot) = -w(\cdot, \cdot)$. To solve this bipartite matching problem, we employ the Minimum Cost Maximum Flow algorithm [1, 37] to find $K$ edges with the minimum cost $w$. The entire curriculum RL process is shown in Algorithm 2 in Appendix E.

### 4.4 Conditional classifier-based intrinsic reward

As we have trained conditional classifiers and defined the pseudo probability, we can additionally use this value as a shaped intrinsic reward for enabling the agent to solve the uninformed search problem. As the pseudo probability outputs 0 for the state in the replay buffer $\mathcal{B}$, 1 for the desired outcome state, and a value between 0 and 1 for the state where the classifiers disagree (meaning unexplored region), we can define the goal-conditioned intrinsic reward as $r = p_{\mathrm{pseudo}}(y = 1|s; g)$. We use this intrinsic reward in all of our experiments. The ablation study for this reward is detailed in Section 5.2.

Table 1: Comparison of our work with the prior curriculum RL methods in conceptual aspects.

| | Conditional quantification | Target dist. of curriculum | Arbitrary desired outcome dist. | Geometry-agnostic | Without external reward |
|---|:---:|:---:|:---:|:---:|:---:|
| HGG | ✗ | $p^+(g)$ | ✓ | ✗ | ✗ |
| CURROT | ✗ | uniform or $p^+(g)$ | ✓ | ✗ | ✗ |
| PLR | ✗ | ✗ | ✗ | ✓ | ✗ |
| VDS | ✗ | ✗ | ✗ | ✓ | ✗ |
| ALP-GMM | ✗ | ✗ | ✗ | ✓ | ✗ |
| OUTPACE | ✗ | $p^+(g)$ | ✗ | ✓ | ✓ |
| **Ours** | ✓ | $p^+(g)$ | ✓ | ✓ | ✓ |

(a) Ours       (b) OUTPACE       (c) HGG

Figure 3: Curriculum goal visualization of the proposed method and baselines. **First row**: Ant Locomotion, **Second row**: Spiral-Maze.

## 5 Experiment

We conduct experiments on 6 environments that have multi-modal desired outcome distribution to validate our proposed method. We use various maze environments (Point Complex-Maze, Medium-Maze, Spiral-Maze) to validate our curriculum proposal capability, which is not limited to specific geometries. Additionally, we evaluate our method on more complex dynamics or other domains such as the Ant-Locomotion and Sawyer-Peg Push, Pick&Place with obstacle environments to demonstrate its effectiveness in domains beyond the navigation. (Refer to Appendix D for more details.)

We compare our method with several previous curriculum generation approaches, each of which possesses the following characteristics. **OUTPACE** [5] prioritizes goals that are considered uncertain and temporally distant from the initial state distribution through meta-learning-based uncertainty quantification and Wasserstein-distance-based temporal distance approximation. **HGG** [37] aims to reduce the distance between the desired outcome state and curriculum distributions, using a value function bias and the Euclidean distance metric. **CURROT** [18] interpolates between the desired outcome state and curriculum distribution while considering the agent's current capability using the Wasserstein distance. **VDS** [49] proposes epistemic uncertainty-based goals by utilizing the value function ensembles. **ALP-GMM** [36] models absolute learning progress score by a GMM. **PLR** [16] prioritizes increasingly challenging tasks by ranking task levels based on TD errors. We summarize the conceptual comparison between our method and baselines in Table 1.

### 5.1 Experimental results

First, to validate the quality of curriculum goal interpolation from the initial state distribution to the desired outcome distribution, we qualitatively and quantitatively evaluate the curriculum progress achieved during the training by optimizing Eq (6). For qualitative evaluation, we visualize the curriculum goals obtained by the proposed method and other baselines (Figure 3). The proposed method shows calibrated guidance toward the desired outcome states, even with the multi-modal distribution. But, HGG shows an inappropriate curriculum proposal due to the Euclidean distance metric, and OUTPACE shows curriculum goals collapsed only toward a single direction because it utilizes the Wasserstein distance for biasing curriculum goals into the temporally distant region. Once the agent explores a temporally far region in a specific direction earlier, the curriculum proposal is collapsed toward this area. In contrast, our method does not have such dependence, which enables

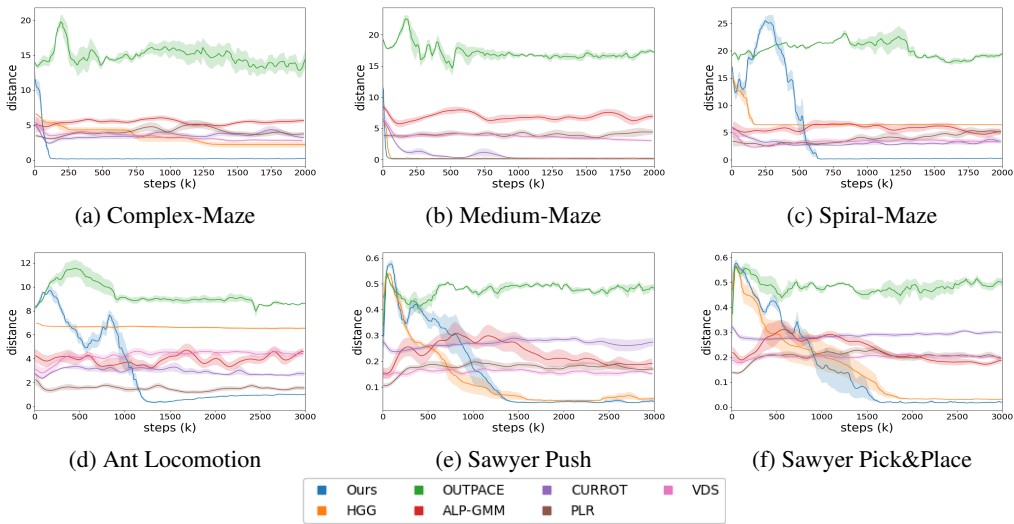

Figure 4: The mean distance between the curriculum goals and final goals (**Lower is better**). The shaded area represents a standard deviation across 5 seeds. The increases of our method at initial steps in some environments are attributed to the geometry of the environments.

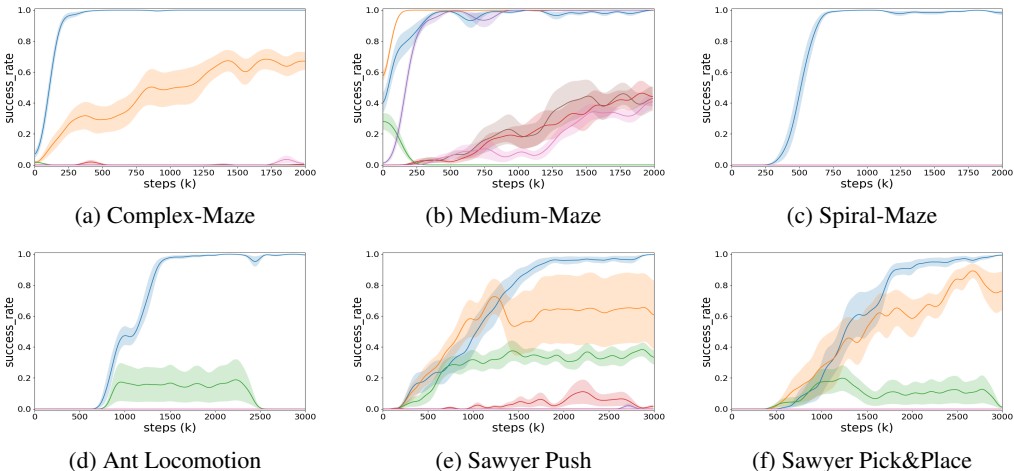

Figure 5: Evaluation success rates, using the same seeds and legends as shown in Figure 4. Note that some baselines are not visible as they coincide with a success rate of zero.

our method to always propose curriculum goals in any uncertain direction quantified by disagreement between the diversified classifiers.

We also plot the average distance from the proposed curriculum goals to the desired outcome states for quantitative evaluation (Figure 4). As we evaluated with multi-modal desired outcome distribution, the distance cannot be measured by naively averaging the distance between the randomly chosen desired outcome states and curriculum goals. Thus, we measure the distance by bipartite matching with the $l_2$ distance metric, which computes the distance between the desired outcome states and their assigned curriculum goals. As shown in Figure 4, only the proposed method consistently shows superior interpolation results from the initial state distribution to desired outcome distribution, while other baselines show some progress only in a simple geometric structure due to the Euclidean distance metric, or get stuck in some local optimum area due to the lack of or insufficient unexplored region quantification. We also plot the performance of the outcome-directed RL in Figure 5. As expected by the average distance measurement in Figure 4, the proposed method is the only one that consistently and quickly achieves the desired outcome states through the guidance of calibrated curriculum goals, indicating the advantage of the proposed diversified conditional classifiers.

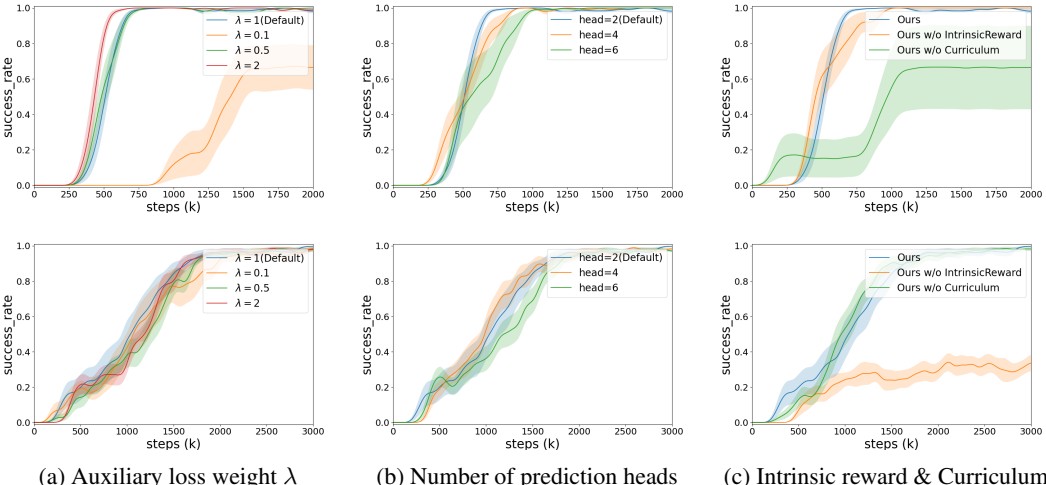

|   |   |   |
|---|---|---|
| (a) Auxiliary loss weight $\lambda$ | (b) Number of prediction heads | (c) Intrinsic reward & Curriculum |

Figure 6: Ablation study in terms of the episode success rates. **First row**: Spiral-Maze. **Second row**: Sawyer Push. The shaded area represents the standard deviation across 5 seeds.

## 5.2 Ablation study

**Number of prediction heads & Auxiliary loss weight $\lambda$.** To investigate the sensitivity to the hyperparameters of our method, we experimented with the different values of the auxiliary loss' weight $\lambda$ and with the different number of prediction heads of the conditional classifiers. As shown in Figure 6a, 6b, the results are slightly dependent on the environment's characteristics such as the presence of object interaction or complex geometry. However, the overall performance trend is not significantly affected by the number of prediction heads and $\lambda$, except in the extreme case, which supports the superiority of our proposed method. More qualitative/quantitative analyses and other ablation studies are included in Appendix F.

**Reward type & Curriculum.** We evaluate the effectiveness of the proposed intrinsic reward and curriculum proposal by conducting two ablation experiments. First, we replace the intrinsic reward with a sparse reward (**Ours w/o IntrinsicReward**), which is typically used in goal-conditioned RL problems. Second, we conduct experiments without the curriculum proposal while utilizing the intrinsic reward (**Ours w/o Curriculum**). As shown in Figure 6c, there is performance degradation without using the proposed intrinsic reward in an environment with complex dynamics since an informative reward signal is crucial. In addition, the absence of a curriculum proposal leads to performance degradation in an environment with complex geometry as it requires carefully crafted guidance for exploration. These results highlight the importance of both proposed components, i.e. intrinsic reward and curriculum proposal.

## 6 Conclusion

We propose **D2C** that 1) performs a classifier diversification process to distinguish the unexplored region from the explored area and desired outcome example and 2) conquers the unexplored region by proposing the curriculum goal and the shaped intrinsic reward. It enables the agent to automatically progress toward the desired outcome states without prior knowledge of the environment. We demonstrate that our method outperforms the previous methods in terms of sample efficiency and geometry-agnostic, desired-outcome-distribution-agnostic curriculum progress, both quantitatively and qualitatively.

**Limitation & Broader impacts.** Despite the promising results, there is a limitation in the scalability of the proposed method since we use small noise to augment the conditioned goal for numerical stability, making it difficult to scale to high-dimensional inputs such as images. Thus, addressing this point would be an interesting future research direction to develop a more generally applicable method. Also, our work is subject to the potential negative societal impacts of RL, but we do not expect to encounter any additional negative impacts specific to this work.

# 7  Acknowledgement

This work was supported by Korea Research Institute for defense Technology Planning and advancement (KRIT) Grant funded by Defense Acquisition Program Administration(DAPA) (No. KRIT-CT-23-003, Development of AI researchers based on deep reinforcement learning and establishment of virtual combat experiment environment).

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

# A  Training & Experiments details

## A.1  Training details

**Baselines.**    The baseline curriculum RL algorithms are trained as follows,

- OUTPACE [5]: We follow the default setting in the original implementation from `https://github.com/jayLEE0301/outpace_official`.
- HGG [37] : We follow the default setting in the original implementation from `https://github.com/Stilwell-Git/Hindsight-Goal-Generation`.
- CURROT [18]: We follow the default setting in the original implementation from `https://github.com/psclklnk/currot`.
- PLR [16], VDS [49], ALP-GMM [36] : We follow the default setting in implementation from `https://github.com/psclklnk/currot`.

D2C and all the baselines are trained by SAC [13] with the sparse reward except for the OUTPACE which uses an intrinsic reward based on Wasserstein distance with a time-step metric.

**Training details.**    We used NVIDIA A5000 GPU and AMD Ryzen Threadripper 3960X for training, and each experiment took about 1∼2 days for training. We used small noise from a uniform distribution with an environment-specific noise scale (Table 5) for augmenting the conditioned goal in Eq (5). Also, we used the mapping $\phi(\cdot)$ that abstracts the state space into the goal space when we use the diversified conditional classifiers (i.e. $f_i(\phi(s); g)$. For example, $\phi(\cdot)$ abstracts the proprioceptive states (e.g. $xy$ position of the agent) in navigation tasks, and abstracts the object-centric states (e.g. $xyz$ position of the object) in robotic manipulation tasks.

Table 2: Hyperparameters for D2C

| critic hidden dim | 512 | discount factor $\gamma$ | 0.99 |
|---|---|---|---|
| critic hidden depth | 3 | batch size | 512 |
| critic target $\tau$ | 0.01 | init temperature $\alpha_{\text{init}}$ of SAC | 0.3 |
| critic target update frequency | 2 | replay buffer $\mathcal{B}$ size | 3e6 |
| actor hidden dim | 512 | learning rate for $f_i$ | 1e-3 |
| actor hidden depth | 3 | learning rate for Critic & Actor | 1e-4 |
| actor update frequency | 2 | optimizer | adam |

Table 3: Default env-specific hyperparameters for D2C

| Env name | # of heads | $\lambda$ | $\epsilon$ | $f_i$ update freq (step) | $f_i$ # of iteration per update | max episode horizon |
|---|---|---|---|---|---|---|
| Complex-Maze | 2 | 1 | 0.5 | 2000 | 16 | 100 |
| Medium-Maze | 2 | 1 | 0.5 | 2000 | 16 | 100 |
| Spiral-Maze | 2 | 1 | 0.5 | 2000 | 16 | 100 |
| Ant Locomotion | 2 | 2 | 1.0 | 4500 | 16 | 300 |
| Sawyer-Peg-Push | 2 | 1 | 0.025 | 3000 | 16 | 200 |
| Sawyer-Peg-Pick&Place | 2 | 1 | 0.025 | 3000 | 16 | 200 |

## A.2  Environment details

- Complex-Maze: The observation consists of the $xy$ position, angle, velocity, and angular velocity of the 'point'. The action space consists of the velocity and angular velocity of the 'point'. The initial state of the agent is $[0, 0]$ and the desired outcome states are obtained from the default goal points $[8, 16], [-8, -16], [16, -8], [-16, 8]$. The size of the map is $36 \times 36$.
- Medium-Maze:   It is the same as the Complex-Maze environment except that the desired outcome states are obtained from the default goal points $[16, 16], [-16, -16], [16, -16], [-16, 16]$.

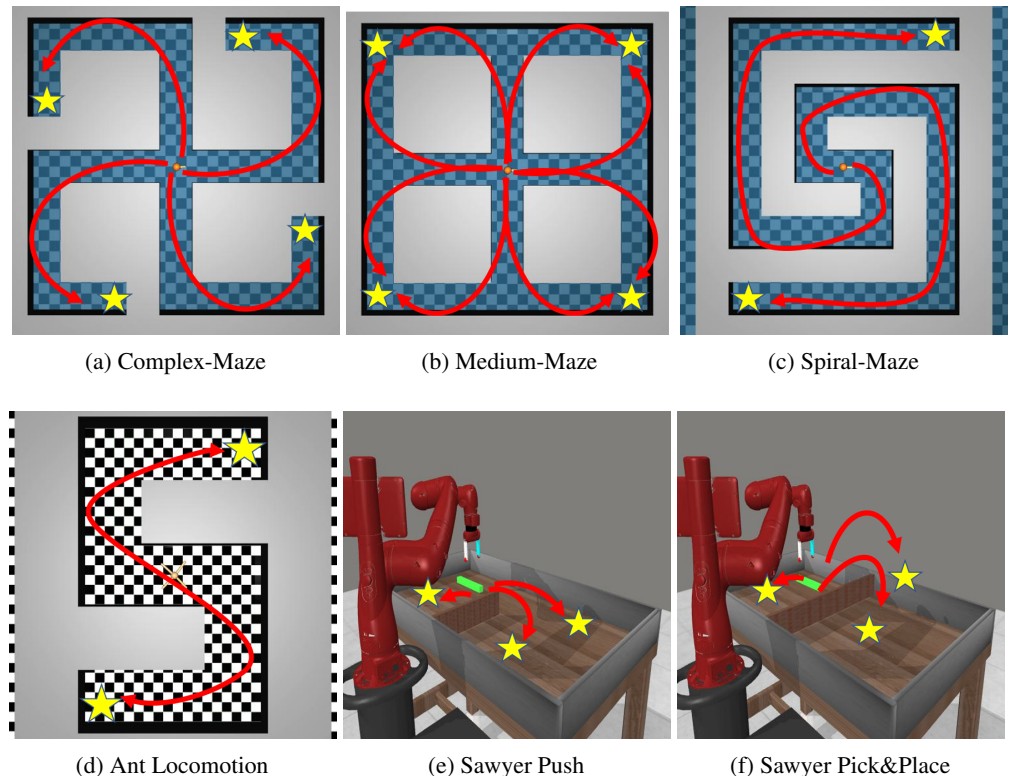

|                       |                    |                    |
|:---------------------:|:------------------:|:------------------:|
| (a) Complex-Maze      | (b) Medium-Maze    | (c) Spiral-Maze    |
| (d) Ant Locomotion    | (e) Sawyer Push    | (f) Sawyer Pick&Place |

Figure 7: Environments used for evaluation: yellow stars indicate the desired outcome examples. **(a)-(c)** the agent should navigate various maze environments with multi-modal desired outcome distribution. **(d)** the ant locomotion environment with multi-modal desired outcome distribution. **(e)** the robot has to push or pick & place a peg to the multi-modal desired locations while avoiding an obstacle at the center of the table.

- Spiral-Maze: The observation space and actions space and initial state of the agent are the same as in the Complex-Maze environment. The desired outcome states are obtained from the default goal points $[12, 16], [-12, -16]$. The size of the map is $28 \times 36$.

- Ant Locomotion: The observation consists of the $xyz$ position, $xyz$ velocity, joint angle, and joint angular velocity of the 'ant'. The action space consists of the torque applied on the rotor of the 'ant'. The initial state of the agent is $[0, 0]$ and the desired outcome states are obtained from the default goal points $[4, 8], [-4, -8]$. The size of the map is $12 \times 20$.

- Sawyer-Peg-Push: The observation consists of the $xyz$ position of the end-effector, the object, and the gripper's state. The action space consists of the $xyz$ position of the end-effector and gripper open/close control. The initial state of the object is $[0.4, 0.8, 0.02]$ and the desired outcome states are obtained from the default goal points $[-0.3, 0.4, 0.02], [-0.3, 0.8, 0.02], [0.4, 0.4, 0.02]$. The wall is located at the center of the table. Thus, the robot arm should detour the wall to reach the desired goal states. We referred to the metaworld [48] and EARL [40] environments.

- Sawyer-Peg-Pick&Place: It is the same as the Sawyer-Peg-Push environment except that the desired outcome states are obtained from the default goal points $[-0.3, 0.4, 0.2], [-0.3, 0.8, 0.2], [0.4, 0.4, 0.2]$, and the wall is located at the center of the table, fully blocking a path for pushing. Thus, the robot arm should pick and move the object over the wall to reach the desired goal states.

# B Algorithm

---

**Algorithm 1** Overview of D2C algorithm

---

1: **Input:** desired outcome examples $\hat{p}^+(g)$, total training episodes $N$, Env, environment horizon $H$, actor $\pi$, critic $Q$, replay buffer $\mathcal{B}$
2: **for** iteration=1,2,...,N **do**
3:    $\hat{p}^c \leftarrow$ sample K curriculum goals that minimize Eq (6). We refer to HGG [37] for solving the bipartite matching problem.
4:    **for** $i$=1,2,...,K **do**
5:       Env.reset()
6:       $g \leftarrow \hat{p}^c$
7:       **for** $t$=0,1,...,$H$-1 **do**
8:          **if** achieved $g$ **then**
9:             $g \leftarrow$ random goal (randomly sample a few states near $s_t$ and measure $p_{\text{pseudo}}$. Then select a state with the highest value of $p_{\text{pseudo}}$.)
10:          **end if**
11:          $a_t \leftarrow \pi(\cdot|s_t, g)$
12:          $s_{t+1} \leftarrow$ Env.step($a_t$)
13:       **end for**
14:       $\mathcal{B} \leftarrow \mathcal{B} \cup \{s_0, a_0, g, s_1...\}$
15:    **end for**
16:    **for** $i$=0,1,...,M **do**
17:       Sample a minibatch b from $\mathcal{B}$ and replace the original reward with intrinsic reward in Section 4.4 (We used relabeling technique based on [2]).
18:       Train $\pi$ and $Q$ with b via SAC [13].
19:       Sample another minibatch b′ from $\mathcal{D}_\text{S} \sim \{(\mathcal{B}, y = 0), (\mathcal{D}_\text{G}, y = 1)\}$ and $\mathcal{D}_\text{T} \sim \mathcal{U}$.
20:       Train $f_i$ with b′ via Eq. (5)
21:    **end for**
22: **end for**

---

# C  More experimental results

## C.1  Full results of the main script

We included the full results of the main script in this section. We include the visualization of the proposed curriculum goals in all environments in Figure 16. The visualization results of the Sawyer-Peg-Pick&Place are not included as it shares the same map with the Sawyer-Peg-Push environment.

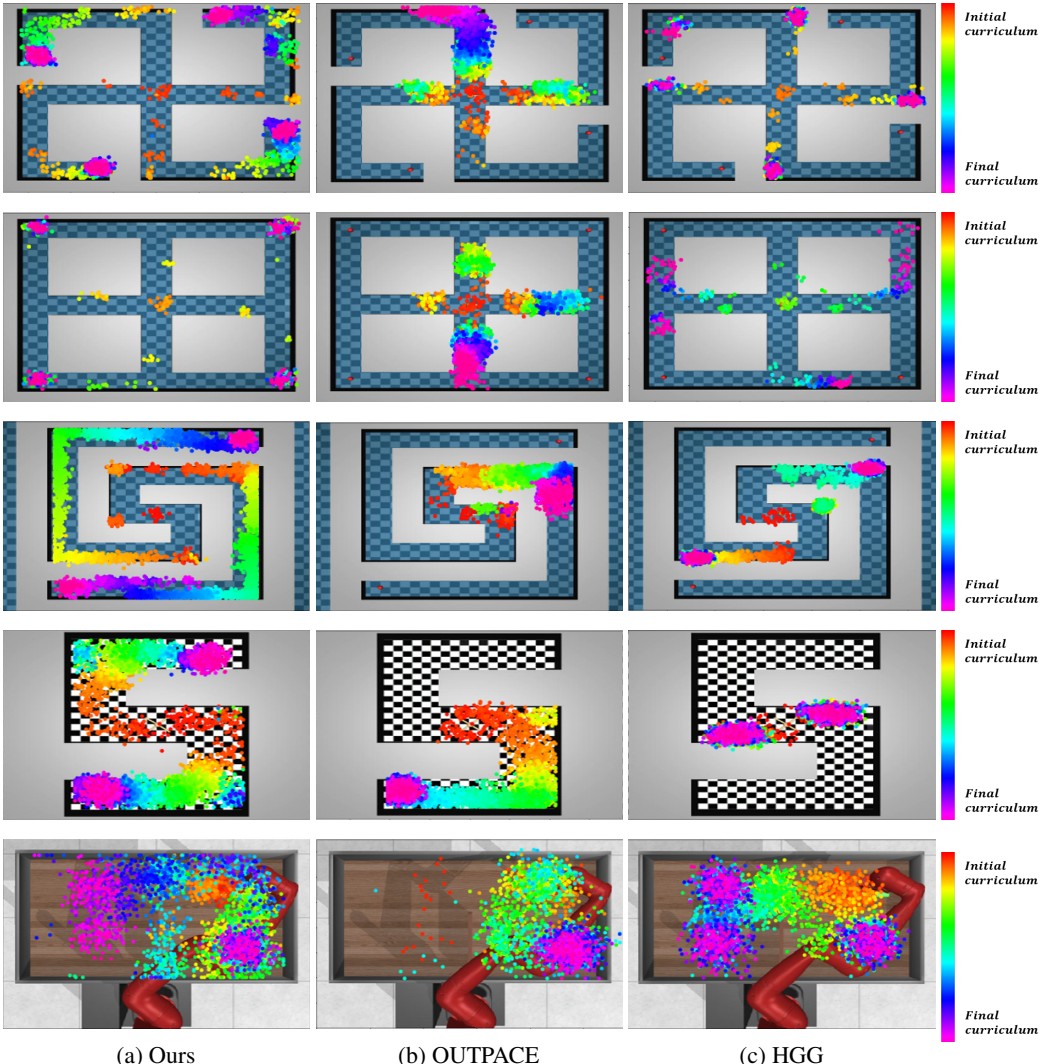

(a) Ours          (b) OUTPACE          (c) HGG

Figure 8: Curriculum goal visualization of the proposed method and baselines in all environments. **First row**: Complex-Maze. **Second row**: Medium-Maze. **Third row**: Spiral-Maze. **Fourth row**: Ant Locomotion. **Fifth row**: Sawyer Push. **Sixth row**: Sawyer Pick & Place.

## C.2 Additional ablation study results

**Full ablation study results of the main script.** We conducted ablation studies described in our main script in all environments. Figure 17 shows the average distance from the proposed curriculum goals to the desired final goal states along the training steps, and Figure 18 shows the episode success rates along the training steps. Note that there are no results for the ablation study without a curriculum proposal in Figure 17c (unlike Figure 18c) since there are no curriculum goals to measure the distance from the desired final goal states. As we can see in these figures, we could obtain consistent analysis with the results in the main script in most of the environments.

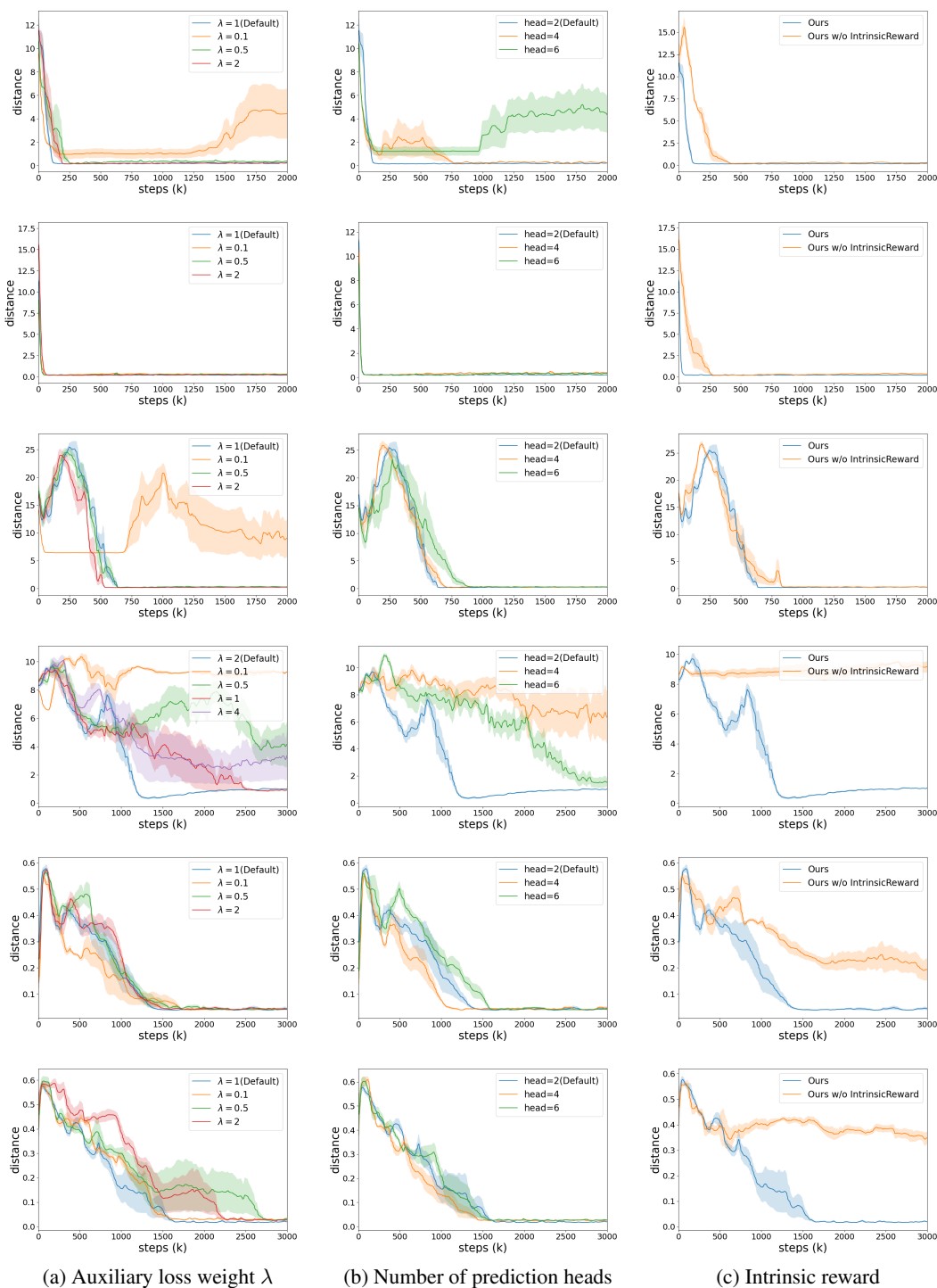

(a) Auxiliary loss weight $\lambda$      (b) Number of prediction heads      (c) Intrinsic reward

Figure 9: Ablation study in terms of the distance from the proposed curriculum goals to the desired final goal states (**Lower is better**). **First row**: Complex-Maze. **Second row**: Medium-Maze. **Third row**: Spiral-Maze. **Fourth row**: Ant Locomotion. **Fifth row**: Sawyer Push. **Sixth row**: Sawyer Pick & Place. The shaded area represents a standard deviation across 5 seeds.

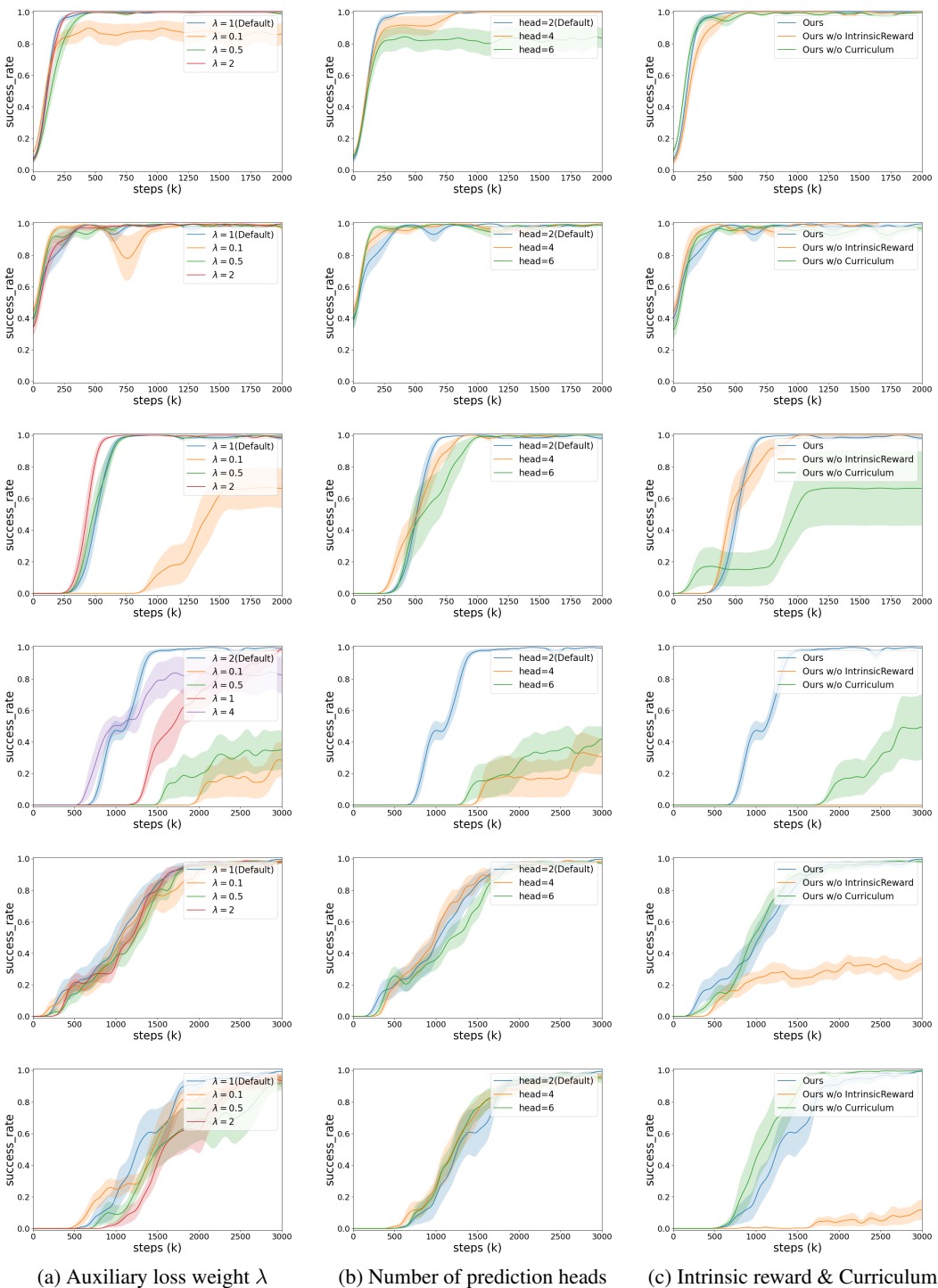

(a) Auxiliary loss weight $\lambda$     (b) Number of prediction heads     (c) Intrinsic reward & Curriculum

Figure 10: Ablation study in terms of the episode success rate. **First row**: Complex-Maze. **Second row**: Medium-Maze. **Third row**: Spiral-Maze. **Fourth row**: Ant Locomotion. **Fifth row**: Sawyer Push. **Sixth row**: Sawyer Pick & Place. The shaded area represents a standard deviation across 5 seeds.

**Curriculum learning objective type.** We conduct additional experiments to validate whether reflecting the temporal distance in a curriculum learning objective (Eq (7)) is required since there are a few works that estimate the temporal distance from the initial state distribution to propose the curriculum goals in a temporally distant region or explore based on this temporal information [5, 37]. To reflect the temporal distance in the cost function (Eq (7)), we modify it as $w(s_i, g_i^+) := \mathcal{CE}(p_{\text{pseudo}}(y = 1|s_i; g_i^+); y = p_{\text{pseudo}}(y = 1|g_i^+; g_i^+)) - V^\pi(s_0, \phi(s_i))$ ($\phi(\cdot)$ is goal space mapping) since the value function itself implicitly represents the temporal distance if we use the sparse reward or custom-defined reward similar to the sparse one. In this case, our proposed intrinsic reward outputs 1 for the desired goal and 0 for the explored states, and it works similarly to the sparse one.

We experimented with this modified curriculum learning objective (**+Value**), and the results are shown in Figure 19, 20. It shows that there is no significant difference, which supports the superiority of our method in that our method achieves state-of-the-art results without considering additional temporal distance information.

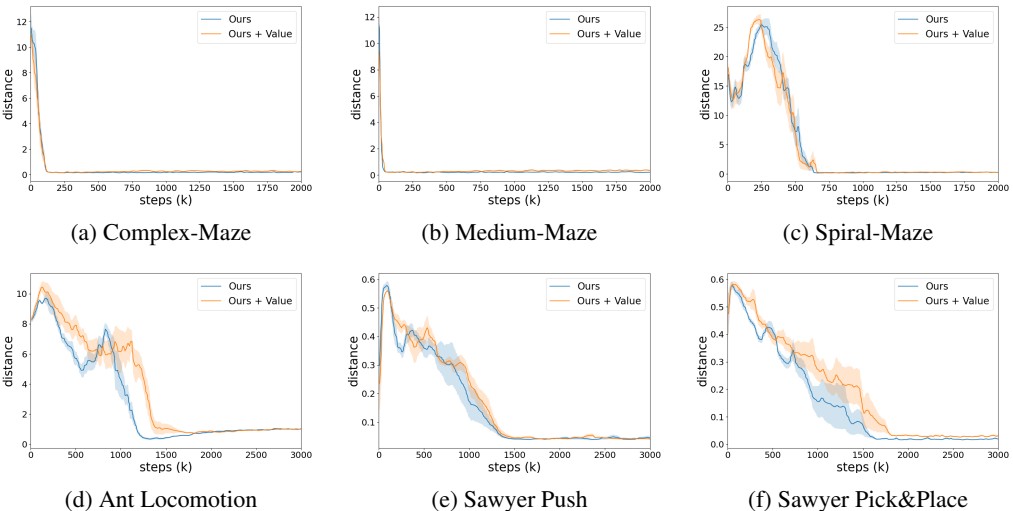

Figure 11: Ablation study in terms of the average distance from the curriculum goals to the final goals (**Lower is better**). +Value means that we additionally consider the value function bias in the curriculum learning objective to reflect the temporal distance from the initial state distribution.

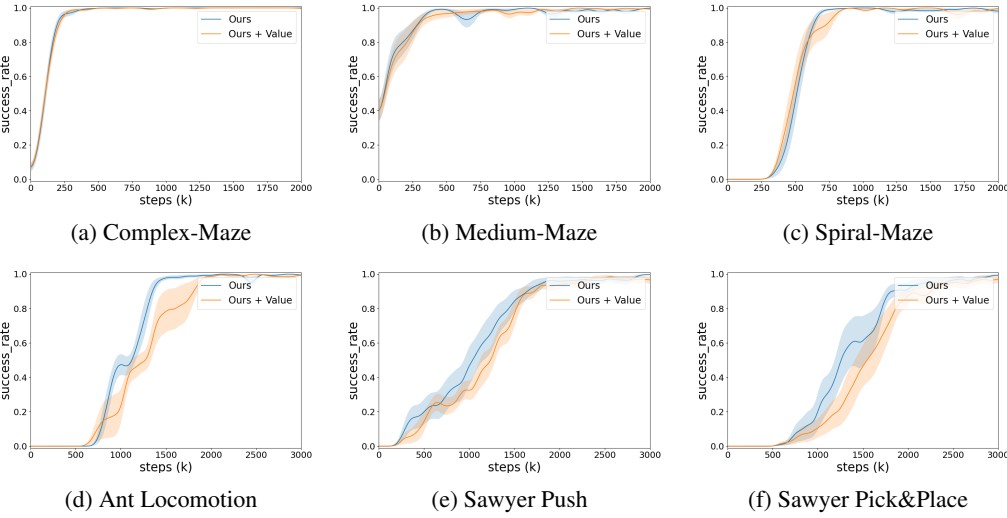

Figure 12: Ablation study in terms of the episode success rates. +Value means that we additionally consider the value function bias in the curriculum learning objective to reflect the temporal distance from the initial state distribution.

**Choice of goal candidates in training conditional classifiers.** As mentioned in the main script, we also experimented with different choices of the goal candidates when we train the conditional classifiers (Eq (5)). The default setting is $\mathcal{D}_G = \mathcal{D}_T$, and we also experimented with $\mathcal{D}_G = \mathcal{B} \cup p^+(g)$. The results are shown in Figure 21, 22. It shows that there is no significant difference, which means we can even make the problem setting more strict by conditioning the classifier only with the visited states and the given desired outcome examples.

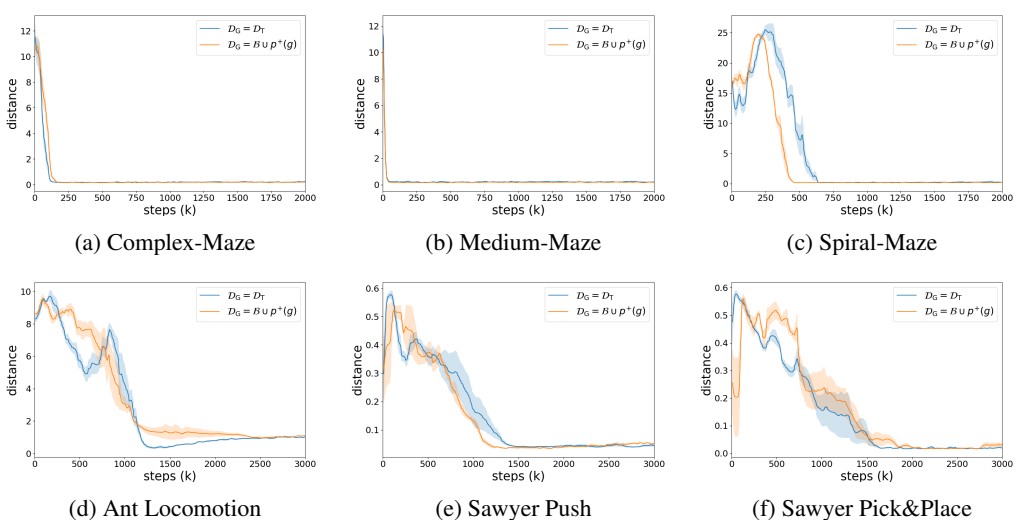

Figure 13: Ablation study in terms of the average distance from the curriculum goals to the final goals (**Lower is better**). There are no significant differences between the choice of goal candidates to train the conditional classifiers.

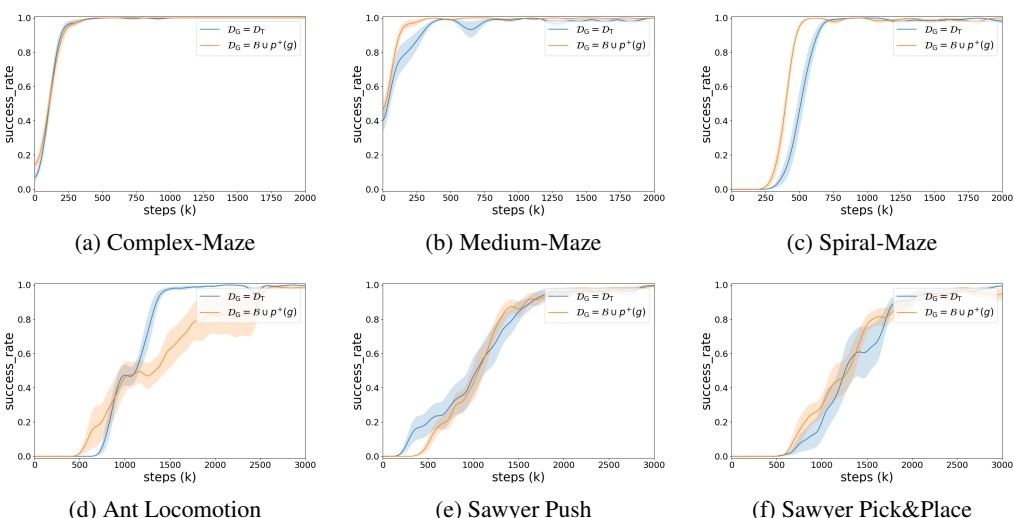

Figure 14: Ablation study in terms of the episode success rates. There are no significant differences between the choice of goal candidates to train the conditional classifiers.

## D  Training & Experiments details

### D.1  Training details

**Baselines.** The baseline curriculum RL algorithms are trained as follows,

- OUTPACE [5]: We follow the default setting in the original implementation from `https://github.com/jayLEE0301/outpace_official`.

- HGG [37] : We follow the default setting in the original implementation from `https://github.com/Stilwell-Git/Hindsight-Goal-Generation`.

- CURROT [18]: We follow the default setting in the original implementation from `https://github.com/psclklnk/currot`.

- PLR [16], VDS [49], ALP-GMM [36] : We follow the default setting in implementation from `https://github.com/psclklnk/currot`.

D2C and all the baselines are trained by SAC [13] with the sparse reward except for the OUTPACE which uses an intrinsic reward based on Wasserstein distance with a time-step metric.

**Training details.** We used NVIDIA A5000 GPU and AMD Ryzen Threadripper 3960X for training, and each experiment took about 1∼2 days for training. We used small noise from a uniform distribution with an environment-specific noise scale (Table 5) for augmenting the conditioned goal in Eq (5). Also, we used the mapping $\phi(\cdot)$ that abstracts the state space into the goal space when we use the diversified conditional classifiers (i.e. $f_i(\phi(s); g)$. For example, $\phi(\cdot)$ abstracts the proprioceptive states (e.g. $xy$ position of the agent) in navigation tasks, and abstracts the object-centric states (e.g. $xyz$ position of the object) in robotic manipulation tasks.

Table 4: Hyperparameters for D2C

| critic hidden dim | 512 | discount factor $\gamma$ | 0.99 |
|---|---|---|---|
| critic hidden depth | 3 | batch size | 512 |
| critic target $\tau$ | 0.01 | init temperature $\alpha_{\text{init}}$ of SAC | 0.3 |
| Critic target update frequency | 2 | replay buffer $\mathcal{B}$ size | 3e6 |
| actor hidden dim | 512 | learning rate for $f_i$ | 1e-3 |
| actor hidden depth | 3 | learning rate for Critic & Actor | 1e-4 |
| actor update frequency | 2 | optimizer | adam |

Table 5: Default env-specific hyperparameters for D2C

| Env name | # of heads | $\lambda$ | $\epsilon$ | $f_i$ update freq (step) | $f_i$ # of iteration per update | max episode horizon |
|---|---|---|---|---|---|---|
| Complex-Maze | 2 | 1 | 0.5 | 2000 | 16 | 100 |
| Medium-Maze | 2 | 1 | 0.5 | 2000 | 16 | 100 |
| Spiral-Maze | 2 | 1 | 0.5 | 2000 | 16 | 100 |
| Ant Locomotion | 2 | 2 | 1.0 | 4500 | 16 | 300 |
| Sawyer-Peg-Push | 2 | 1 | 0.025 | 3000 | 16 | 200 |
| Sawyer-Peg-Pick&Place | 2 | 1 | 0.025 | 3000 | 16 | 200 |

## D.2 Environment details

- Complex-Maze: The observation consists of the $xy$ position, angle, velocity, and angular velocity of the 'point'. The action space consists of the velocity and angular velocity of the 'point'. The initial state of the agent is $[0, 0]$ and the desired outcome states are obtained from the default goal points $[8, 16], [-8, -16], [16, -8], [-16, 8]$. The size of the map is $36 \times 36$.

- Medium-Maze: It is the same as the Complex-Maze environment except that the desired outcome states are obtained from the default goal points $[16, 16], [-16, -16], [16, -16], [-16, 16]$.

- Spiral-Maze: The observation space and actions space and initial state of the agent are the same as in the Complex-Maze environment. The desired outcome states are obtained from the default goal points $[12, 16], [-12, -16]$. The size of the map is $28 \times 36$.

- Ant Locomotion: The observation consists of the $xyz$ position, $xyz$ velocity, joint angle, and joint angular velocity of the 'ant'. The action space consists of the torque applied on the rotor of the 'ant'. The initial state of the agent is $[0, 0]$ and the desired outcome states are obtained from the default goal points $[4, 8], [-4, -8]$. The size of the map is $12 \times 20$.

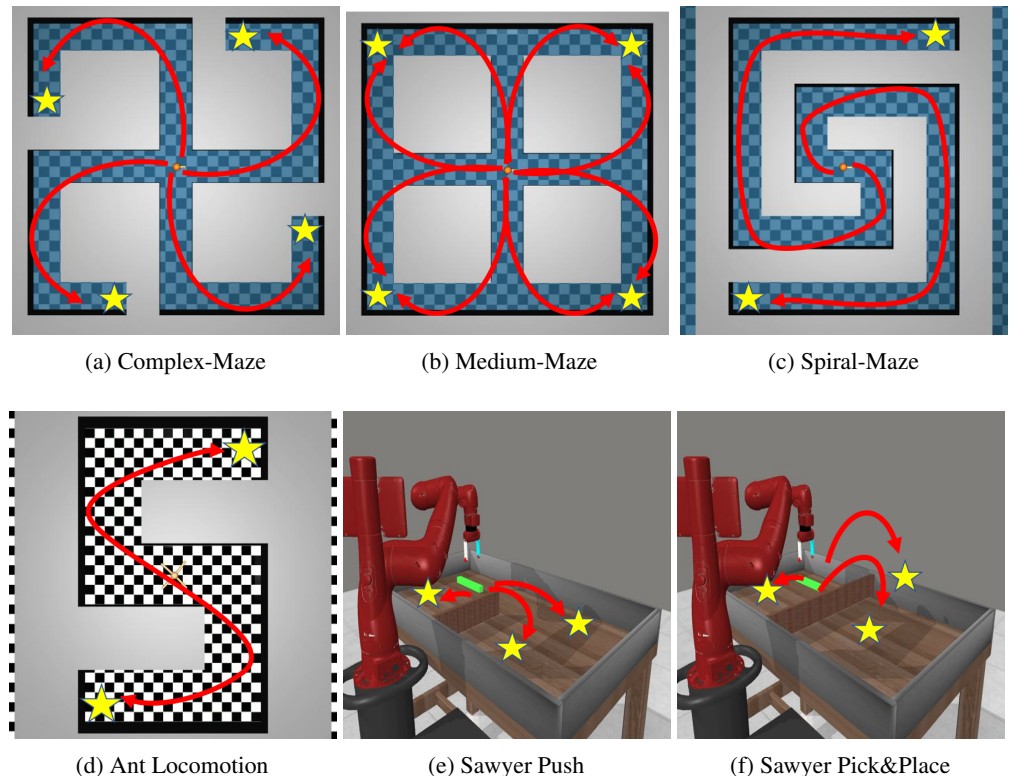

| (a) Complex-Maze | (b) Medium-Maze | (c) Spiral-Maze |
| (d) Ant Locomotion | (e) Sawyer Push | (f) Sawyer Pick&Place |

Figure 15: Environments used for evaluation: yellow stars indicate the desired outcome examples. **(a)-(c)** the agent should navigate various maze environments with multi-modal desired outcome distribution. **(d)** the ant locomotion environment with multi-modal desired outcome distribution. **(e)** the robot has to push or pick & place a peg to the multi-modal desired locations while avoiding an obstacle at the center of the table.

- Sawyer-Peg-Push: The observation consists of the $xyz$ position of the end-effector, the object, and the gripper's state. The action space consists of the $xyz$ position of the end-effector and gripper open/close control. The initial state of the object is $[0.4, 0.8, 0.02]$ and the desired outcome states are obtained from the default goal points $[-0.3, 0.4, 0.02], [-0.3, 0.8, 0.02], [0.4, 0.4, 0.02]$. The wall is located at the center of the table. Thus, the robot arm should detour the wall to reach the desired goal states. We referred to the metaworld [48] and EARL [40] environments.

- Sawyer-Peg-Pick&Place: It is the same as the Sawyer-Peg-Push environment except that the desired outcome states are obtained from the default goal points $[-0.3, 0.4, 0.2], [-0.3, 0.8, 0.2], [0.4, 0.4, 0.2]$, and the wall is located at the center of the table, fully blocking a path for pushing. Thus, the robot arm should pick and move the object over the wall to reach the desired goal states.

# E   Algorithm

---

**Algorithm 2** Overview of D2C algorithm

---

1: **Input:** desired outcome examples $\hat{p}^+(g)$, total training episodes $N$, Env, environment horizon $H$, actor $\pi$, critic $Q$, replay buffer $\mathcal{B}$
2: **for** iteration=1,2,...,N **do**
3:     $\hat{p}^c \leftarrow$ sample K curriculum goals that minimize Eq (6). We refer to HGG [37] for solving the bipartite matching problem.
4:     **for** $i$=1,2,...,K **do**
5:         Env.reset()
6:         $g \leftarrow \hat{p}^c$
7:         **for** $t$=0,1,...,$H$-1 **do**
8:             **if** achieved $g$ **then**
9:                 $g \leftarrow$ random goal (randomly sample a few states near $s_t$ and measure $p_{\text{pseudo}}$. Then select a state with the highest value of $p_{\text{pseudo}}$.)
10:             **end if**
11:             $a_t \leftarrow \pi(\cdot|s_t, g)$
12:             $s_{t+1} \leftarrow$ Env.step($a_t$)
13:         **end for**
14:         $\mathcal{B} \leftarrow \mathcal{B} \cup \{s_0, a_0, g, s_1...\}$
15:     **end for**
16:     **for** $i$=0,1,...,M **do**
17:         Sample a minibatch b from $\mathcal{B}$ and replace the original reward with intrinsic reward in Section 4.4 (We used relabeling technique based on [2]).
18:         Train $\pi$ and $Q$ with b via SAC [13].
19:         Sample another minibatch b' from $\mathcal{D}_S \sim \{(\mathcal{B}, y = 0), (\mathcal{D}_G, y = 1)\}$ and $\mathcal{D}_T \sim \mathcal{U}$.
20:         Train $f_i$ with b' via Eq. (5)
21:     **end for**
22: **end for**

---

# F  More experimental results

## F.1  Full results of the main script

We included the full results of the main script in this section. We include the visualization of the proposed curriculum goals in all environments in Figure 16. The visualization results of the Sawyer-Peg-Pick&Place are not included as it shares the same map with the Sawyer-Peg-Push environment.

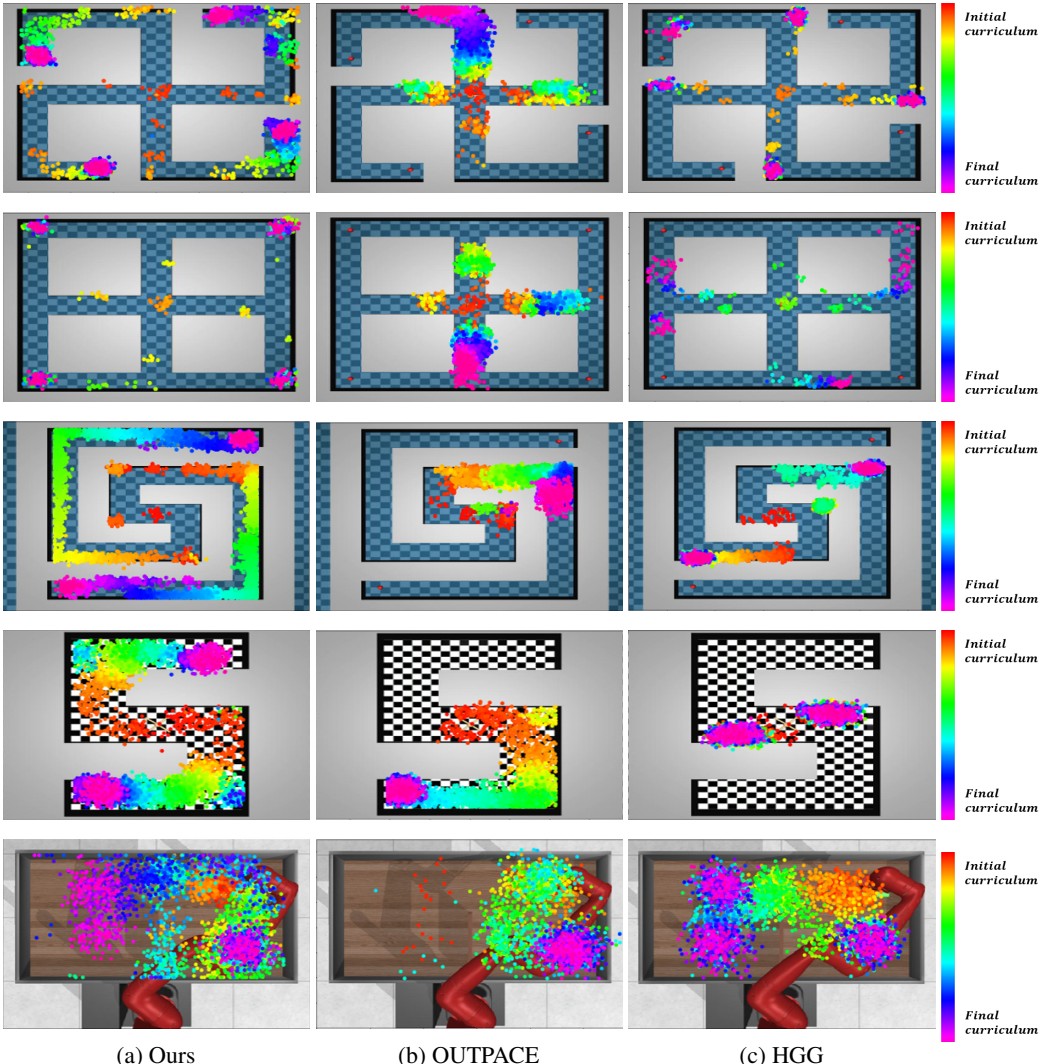

(a) Ours                    (b) OUTPACE                    (c) HGG

Figure 16: Curriculum goal visualization of the proposed method and baselines in all environments. **First row**: Complex-Maze. **Second row**: Medium-Maze. **Third row**: Spiral-Maze. **Fourth row**: Ant Locomotion. **Fifth row**: Sawyer Push. **Sixth row**: Sawyer Pick & Place.

### F.2 Additional ablation study results

**Full ablation study results of the main script.** We conducted ablation studies described in our main script in all environments. Figure 17 shows the average distance from the proposed curriculum goals to the desired final goal states along the training steps, and Figure 18 shows the episode success rates along the training steps. As we can see in these figures, we could obtain consistent analysis with the results in the main script in most of the environments.

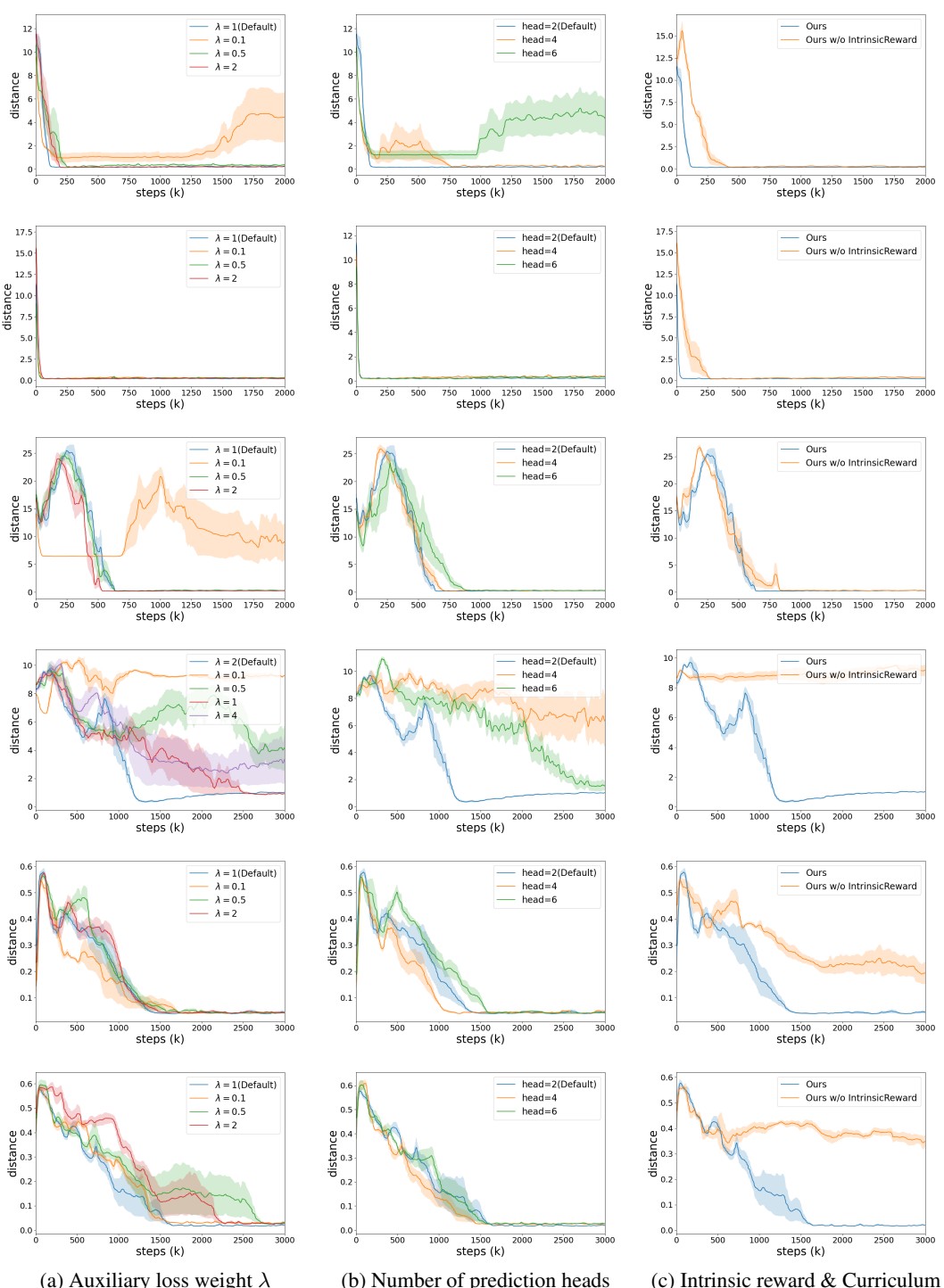

(a) Auxiliary loss weight $\lambda$     (b) Number of prediction heads     (c) Intrinsic reward & Curriculum

Figure 17: Ablation study in terms of the distance from the proposed curriculum goals to the desired final goal states (**Lower is better**). There are no results for the ablation study without a curriculum proposal since there are no curriculum goals to measure the distance from the desired final goal states. **First row**: Complex-Maze. **Second row**: Medium-Maze. **Third row**: Spiral-Maze. **Fourth row**: Ant Locomotion. **Fifth row**: Sawyer Push. **Sixth row**: Sawyer Pick & Place. The shaded area represents a standard deviation across 5 seeds.

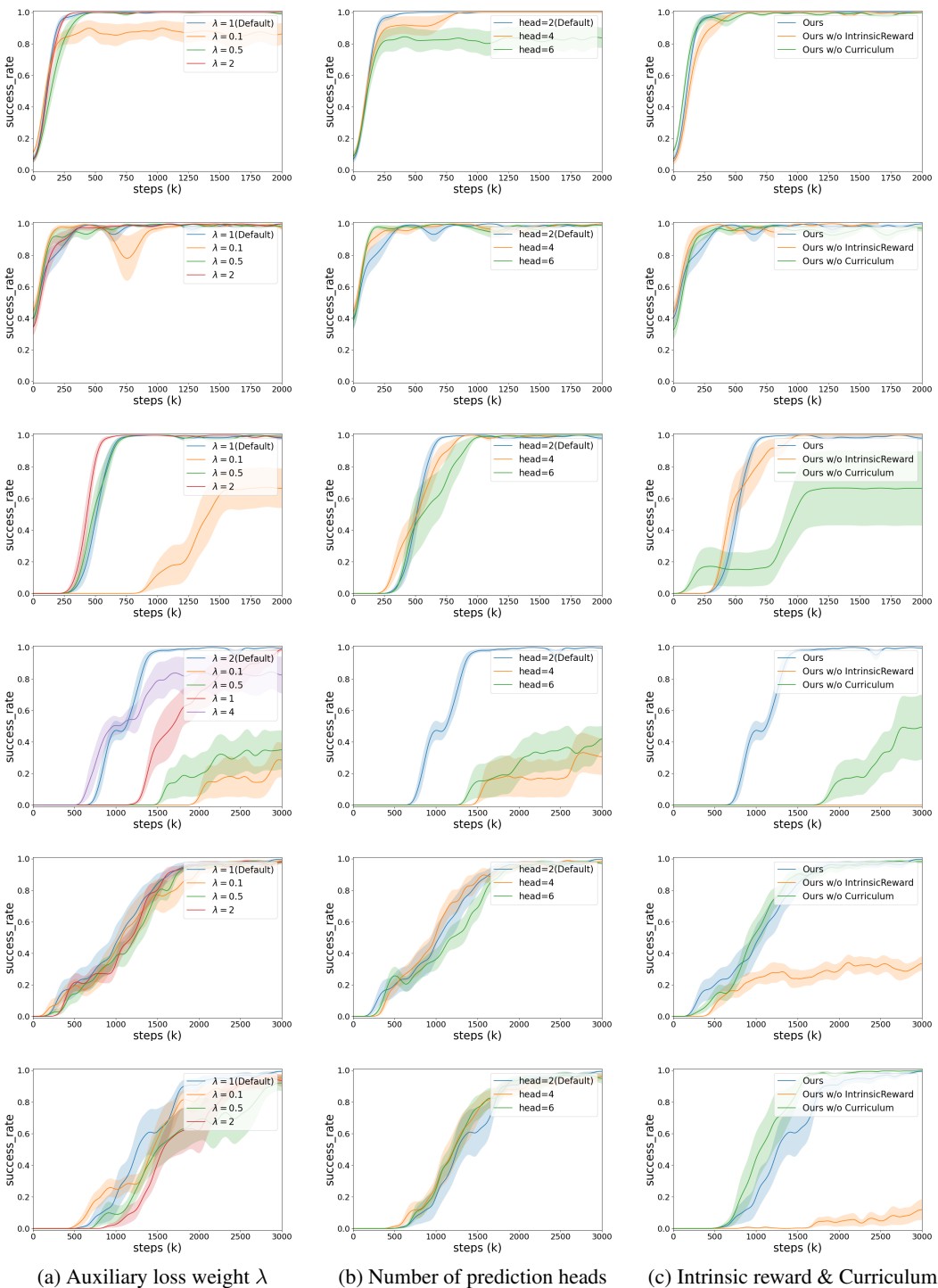

(a) Auxiliary loss weight $\lambda$    (b) Number of prediction heads    (c) Intrinsic reward & Curriculum

Figure 18: Ablation study in terms of the episode success rate. **First row**: Complex-Maze. **Second row**: Medium-Maze. **Third row**: Spiral-Maze. **Fourth row**: Ant Locomotion. **Fifth row**: Sawyer Push. **Sixth row**: Sawyer Pick & Place. The shaded area represents a standard deviation across 5 seeds.

**Curriculum learning objective type.** We conduct additional experiments to validate whether reflecting the temporal distance in a curriculum learning objective (Eq (7)) is required since there are a few works that estimate the temporal distance from the initial state distribution to propose the curriculum goals in a temporally distant region or explore based on this temporal information [5, 37]. To reflect the temporal distance in the cost function (Eq (7)), we modify it as $w(s_i, g_i^+) := \mathcal{CE}(p_{\text{pseudo}}(y = 1|s_i; g_i^+); y = p_{\text{pseudo}}(y = 1|g_i^+; g_i^+)) - V^\pi(s_0, \phi(s_i))$ ($\phi(\cdot)$ is goal space mapping) since the value function itself implicitly represents the temporal distance if we use the sparse reward or custom-defined reward similar to the sparse one. In this case, our proposed intrinsic reward outputs 1 for the desired goal and 0 for the explored states, and it works similarly to the sparse one.

We experimented with this modified curriculum learning objective (**+Value**), and the results are shown in Figure 19, 20. It shows that there is no significant difference, which supports the superiority of our method in that our method achieves state-of-the-art results without considering additional temporal distance information.

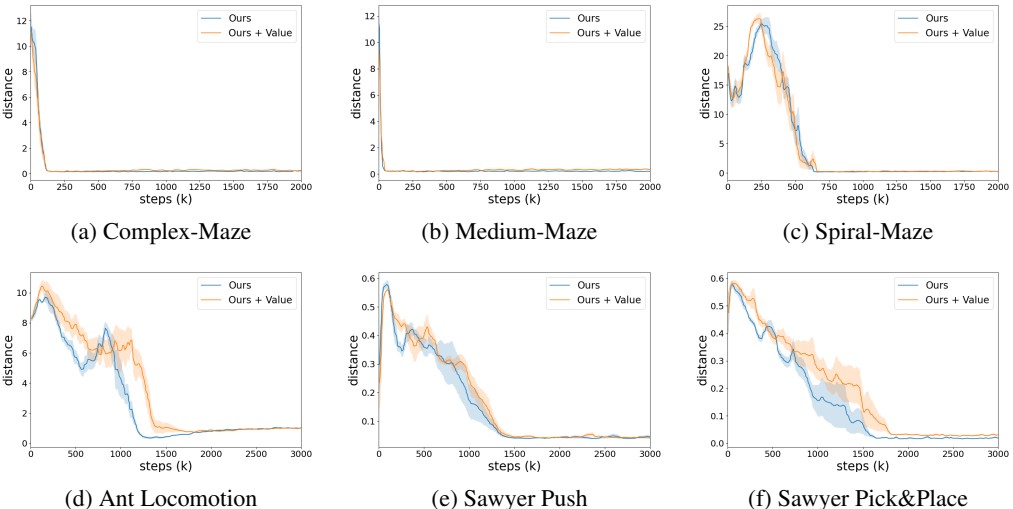

Figure 19: Ablation study in terms of the average distance from the curriculum goals to the final goals (**Lower is better**). +Value means that we additionally consider the value function bias in the curriculum learning objective to reflect the temporal distance from the initial state distribution.

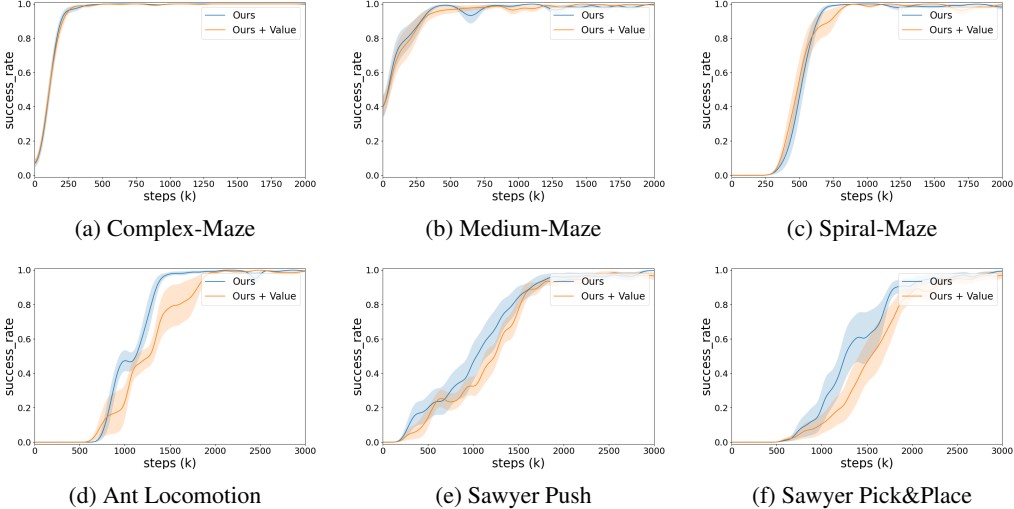

Figure 20: Ablation study in terms of the episode success rates. +Value means that we additionally consider the value function bias in the curriculum learning objective to reflect the temporal distance from the initial state distribution.

**Choice of goal candidates in training conditional classifiers.** As mentioned in the main script, we also experimented with different choices of the goal candidates when we train the conditional classifiers (Eq (5)). The default setting is $\mathcal{D}_G = \mathcal{D}_T$, and we also experimented with $\mathcal{D}_G = \mathcal{B} \cup p^+(g)$. The results are shown in Figure 21, 22. It shows that there is no significant difference, which means we can even make the problem setting more strict by conditioning the classifier only with the visited states and the given desired outcome examples.

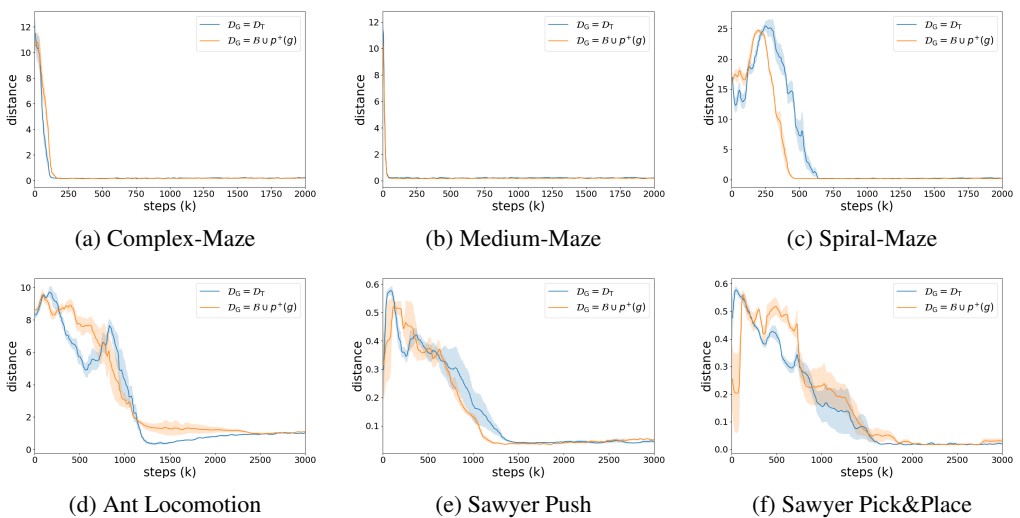

Figure 21: Ablation study in terms of the average distance from the curriculum goals to the final goals (**Lower is better**). There are no significant differences between the choice of goal candidates to train the conditional classifiers.

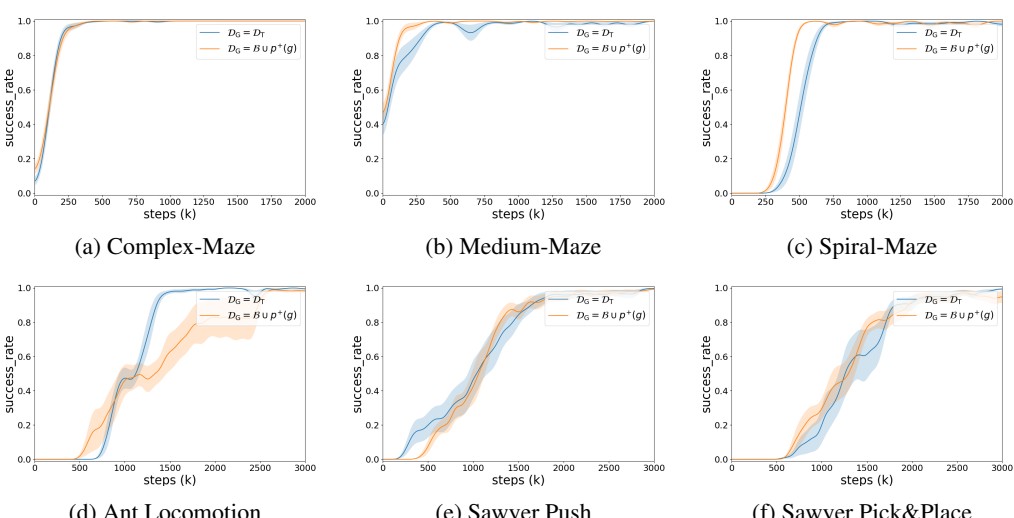

Figure 22: Ablation study in terms of the episode success rates. There are no significant differences between the choice of goal candidates to train the conditional classifiers.

