# OpenReview forum: "Diversify \& Conquer: Outcome-directed Curriculum RL via Out-of-Distribution Disagreement"
_NeurIPS.cc/2023/Conference — NeurIPS 2023 poster_

### Official Review · Reviewer_TCnF · 2023-06-29

**Soundness:** 2 fair
**Presentation:** 2 fair
**Contribution:** 2 fair
**Rating:** 4
**Confidence:** 3

**Summary:**

This paper introduces a novel curriculum learning algorithm (D2C) in the context of reinforcement learning. Unlike previous approaches, D2C does not rely on prior knowledge of the environment's structure, including the distance measure between states. Instead, it leverages a goal-conditioned binary classifier to differentiate between visited states and desired states. When learning this classifier, D2C additionally incorporates a diversification objective to encourage the classifier to disagree on unvisited states. The paper claims that D2C could automatically generate a diverse set of curriculum goals. These goals, in turn, provide intrinsic rewards that guide the agent towards achieving the desired goal state.


**Strengths:**

1. The technique introduced in this paper that identify the similarities between the explored states and desired outcomes with a diversification objective appears to be novel within the domain of curriculum RL.
2. Empirical evaluation of the proposed methods in the selected maze and robotics manipulation environment shows the significant improvement of the proposed algorithm.


**Weaknesses:**

1. The D2C algorithm relies on having access to a dataset of unlabeled target data in order to model unvisited states. However, this assumption can be a significant limitation of the algorithm. In the experiments conducted in this paper, the author was able to overcome this limitation by uniformly sampling states from the upper and lower bounds of the state space. Nevertheless, obtaining such a collection of (legal) states within the state space of agents or robots is often a challenging task. For instance, if the agent receives pixel observations or if the legal state space of a robotic arm is a constrained subset of the entire space defined by the cartesian product each dimension, the assumption of sampling from the legal state space to obtain the unlabeled target dataset may not be applicable. In general, this assumption holds true only in specific environments or tasks.
2. The current version of the proposed algorithm has some fundamental limitations. For example, the proposed algorithm could be prone to over-explore. Also, how the diversification process helps the overall curriculum proposal process is not clear to me. Please refer to my questiona in the next section.

I am happy to raise the score if my questions are addressed by the author.


**Questions:**

1. Regarding the diversification: It seems that when optimizing the objective of curriculum distribution, the uncertainty/disagreement between different classification heads are not in the objective. Equation 4 simply minimizes the cross entropy between the averaged probability of the proposed curriculum state being a goal state and the average probability of the true goal states. Where does the uncertainty of the unexplored region play a role in the overall objective of the algorithm?
2. The problem of over-exploration: Based on the learning objective of D2C, suppose the agent is given a maze environment similar to the one shown in Figure 2. But if the goal distribution only includes the top one, then is it true that this proposed algorithm would still explore both the top and bottom part of the maze?

---

> ### Author Rebuttal · Authors · 2023-08-08
>
> **Weaknesses:**
>
> 1. The D2C algorithm relies on ...
>
> → We appreciate your insightful comment about the extension to the high-dimensional spaces. As you have rightly pointed out, directly applying the bipartite matching problem (section 4.3) in high-dimensional spaces may not work. However, we can consider an encoder-decoder structure with ELBO loss-based representation learning techniques such as VAE to abstract high-dimensional inputs to low-dimensional latent vectors $z$ and measure the Euclidean distance between $z$ of the querying image and goal image similar to [1]. We think these ideas may extend our bidirectional curriculum generation method to high-dimensional spaces and not be restricted to low-dimensional spaces.
>
> Also, for state inputs in the diversified conditional classifiers, we used the mapping $\phi(\cdot)$ that abstracts the state space into the goal space as described in Appendix A (e.g. $\phi(\cdot)$ abstracts the global xyz position of the object in robot manipulation task). It enables us to construct simple and legal state space within the lower and upper bound in most cases. Since knowing each state element’s meaning is a commonly utilized assumption in RL (e.g. hindsight relabeling technique), we think this is not a significant limitation.
>
> Also, as you may concern, there could be infeasible state space within the lower and upper bound. For instance, we can consider the existence of the obstacle at the center of the end-effector’s feasible range, or conflict between the end-effector’s position obtained by applying forward kinematics to the (uniform) randomly sampled joint position and the end-effector’s position (if joint states are included in states). However, the D2C classifiers disagree even in these infeasible states because these are not included both the desired outcome examples and explored states in the replay buffer because the agent cannot reach these states. Since we obtain the curriculum goals from the replay buffer, infeasible states cannot be proposed as curriculum goals, and the agent will keep explore toward feasible, unexplored areas based on the disagreement between classifiers, and discover the desired outcome states.
>
> If you still have remaining concerns or if we misunderstood your question, please let us know. We would be happy to discuss this.
>
> [1] Nair, Ashvin V., et al. "Visual reinforcement learning with imagined goals." *Advances in neural information processing systems*
>      31 (2018).
>
> 2. The current version of the proposed ...
>
> → Question section
>
> **Questions:**
>
> 1. Regarding the diversification ...
>
> → Thank you for your comment about the diversification. The pseudo probability in Eq (3) represents how similar the queried state is to the explored states or desired outcome states (lines 178-188), and disagreement between classification heads is represented as intermediate values between the 0 (explored states) and 1 (desired outcome states) when optimizing the curriculum loss in Eq (4).
>
> For example, before discovering the desired outcome states, the batch data sampled from the replay buffer in Eq (4) include a diverse range of states from frequently visited states to rarely visited ones such as the states in the frontier of the explored region. As described in lines 178-188, $P_{pseudo}(y=1|g^+)$ is nearly 1 and the loss will be large for the frequently visited states as all classifiers correctly classify them as a label 0 (i.e. $P_{pseudo}(y=1|s)=0$), and the loss will be small for the states in the frontier since the classifiers disagree due to the diversification (i.e. $P_{pseudo}(y=1|s)$ is larger than 0). Thus, the curriculum objective encourages selecting the states in the frontier region.
>
> After discovering the desired outcome states, the curriculum objective encourages selecting the states close to the desired outcome state since the corresponding loss will be the smallest among the losses of states in the replay buffer.
>
> For clarity, we would like to note that cross-entropy in Eq (4) is used as a convex function that decreases as $P_{pseudo}(y=1|s)$ increases, not as an exact entropy minimization to match the probability distribution (because $P_{pseudo}$ in Eq (3) is not mathematically probability, so we coined it with “pseudo”).
>
> We believe that these responses together adequately address this comment. However, if you still have remaining concerns or questions please let us know.
>
> 2. The problem of over-exploration ...
>
> → Thank you for your insightful comment. If the goal distribution only includes the top one, then D2C will propose curriculum goals in both directions at the initial phase, since the frontier states of both ways will have the pseudo probability of 0.5 (for example, in the case of two classification heads). However, as the agent discovers the goal distribution at the top, the curriculum goals will be converged to the distribution at the top since states in these distributions will have the pseudo probability of 1, and the agent does not explore toward the other direction anymore.
>
> We would like to note that we assume a setting where only some desired outcome examples are given before training begins and access to the desired outcome distribution is unavailable. Therefore, the setting where the goal distribution only includes the top one means we want the agent to reach the top one only. Thus, after discovering the top one, no more exploration toward the other direction is a natural consequence. If we want to explore the other direction also, it means there exist desired outcomes in the other direction, so we should be equipped with the desired outcome examples corresponding to the bottom one before training begins. The detailed motivation of this outcome/example-driven method is well summarized in prior works such as [1].
>
> [1] Fu, Justin, et al. "Variational inverse control with events: A general framework for data-driven reward definition." *Advances in neural information processing systems* 31 (2018).

---

> > ### Comment · Reviewer_TCnF · 2023-08-16
> > **Response to Authors**
> >
> > Thank you for the detailed response. I've examined other reviews and all of your replies closely. Nonetheless, I feel my concerns haven't been sufficiently addressed and so I will keep my original score.
> >
> > My doubts regarding the strong assumptions of D2C persist, specifically concerning the access to a collection of unlabeled target data/states. It's challenging to envision how this method can scale to pixel observations. For instance, in the sawyer push environment, if the problem is being tackled solely from pixel observations, how would one amass such a target dataset without prior knowledge of the task?
> >
> > Furthermore, I'm skeptical about the real benefits of borrowing the idea of diversification & disambiguation from [1] to RL in additional to this extra assumption. The goal in the original paper is to learn diverse hypotheses so that at least one learned hypothesis/network does not depend on spurious features. However, in the settings we have here, the goal is to encourage the agent to explore unseen states and reach the goal states in the end, which is of a completely different purpose. As pointed out by the reviewer geTA, there are many existing methods proposed as exploration methods for RL, such as RND [2], Plan2Explore [3], Plan2Predict [4], and so on. Although the author comments on the problem of over-exploration induced by these methods, I don't think I would agree with the argument. As long as a large external/sparse reward is assigned for goal achievement, the agent will secure a high reward upon reaching this state. It will learn from the reward feedback and know how to reach the goal state. Hence, I'm afraid I have to disagree with the author's noted limitation, and I am doubtful of the benefits of incorporating diversification & disambiguation into this setting.
> >
> > Reference:
> >
> > [1] Lee et al. Diversify and Disambiguate: Learning From Underspecified Data
> >
> > [2] Burda et al. Exploration by Random Network Distillation.
> >
> > [3] Sekar et al. Planning to Explore via Self-Supervised World Models
> >
> > [4] Wu et al. Plan To Predict: Learning an Uncertainty-Foreseeing Model for Model-Based Reinforcement Learning

---

> > > ### Author Response · Authors · 2023-08-17
> > >
> > > Thank you for your detailed response. As described in the Limitation section, our work in a current form might not be applicable in the pixel observation setting, as you rightly pointed out.
> > >
> > > To alleviate it, we can consider a setting where the agent additionally explores via random action after reaching the proposed goal image and the classifiers do not utilize the random action-based transition data as label 0 data. If we treat all the images in the buffer, desired outcome examples, and transitions obtained via random action as unlabeled target data, then the classifiers will disagree only on the images in randomly explored areas, which enable us to query unseen, frontier sites and propose curriculum goal image at the frontier of the explored region. Although it depends on the random exploration strategies' effectiveness, this could be one of the possible options for the extension of our work to address the pixel-based setting.
> > >
> > > Furthermore, we would like to note that most of the prior works (e.g. OUTPACE, HGG, CURROT, GoalGAN, etc) that explicitly propose curriculum are also not applicable to the pixel-based setting since these have assumptions/limitations similar to our work or cannot propose the curriculum goal image that corresponds to a completely novel state even with the assistance of generative models.
> > >
> > > Also, as described in the response and manuscript, our problem setting is an example/outcome-directed RL where access to the ground truth (external) reward function is unavailable. Since the referred works [2,3,4] can be used only when we can explicitly access the reward function itself or implicitly access this through environmental interaction, our method is completely different from the RL with external+intrinsic reward setting in [2,3,4]. Therefore, in addition to the benefits of our method described in response to the reviewer geTA, we believe that directly contrasting our work with the prior works that require access to the ground truth reward function is neither necessary nor appropriate.
> > >
> > > Of course, we acknowledge that conducting experiments with pixel-based scenarios would enhance the quality of our work. Nevertheless, it is clear that addressing the “without external reward” framework itself is not only necessary but also increasingly crucial to effectively address RL problems that closely resemble real-world scenarios where access to the ground truth reward function is unavailable. In this regard, we wish to emphasize that D2C has contributions in terms of providing a foundation for addressing the example/outcome-directed RL approach with curriculum learning.

---

### Official Review · Reviewer_WkJV · 2023-07-04

**Soundness:** 3 good
**Presentation:** 2 fair
**Contribution:** 2 fair
**Rating:** 7
**Confidence:** 1

**Summary:**

This paper proposes a new reinforcement learning algorithm called Diversify for Disagreement & Conquer (D2C).
The idea of the algorithm is to divide the search space and let conditional classifiers explore these subspaces.
With a set of experiments, the authors verify the effectiveness of D2C compared to various other approaches.
As new to the field, I can not judge the relevance of the contribution for reinforcement learning.

**Strengths:**

As a person new to the field of RF, this paper is well written since it introduces necessary concepts at an appropriate depth, and the idea and motivation are intuitive.
In addition, the approach is geometric flexible, shows its effectiveness in a range of experiments, and the author provides an in-depth ablation study of relevant hyperparameters.


**Weaknesses:**

Again I need to admit that I am entirely new to this field. At the same time, the other baseline experiments all do explore the search undirected. Is it a fair comparison when your approach can explore many directions situationally?


**Questions:**

See above

**Limitations:**

---

> ### Author Rebuttal · Authors · 2023-08-08
>
> **Weaknesses:**
>
> 1. Again I need to admit that I am entirely new to this field. At the same time, the other baseline experiments all do explore the search undirected. Is it a fair comparison when your approach can explore many directions situationally?
>
> → First of all, it is slightly unclear the meaning of “situationally”, so if we misunderstood your question and the answer is not appropriate, please let us know.
>
> We would like to note that most of the baselines and our method perform undirected exploration until the agent discovers the desired outcome states. Only HGG and CURROT perform directed exploration from the initial phase. Some baselines keep doing undirected exploration even after the agent discovers the desired outcome states (e.g. VDS, ALP-GMM, PLR), and the other baselines and our method propose converged curriculum goals after discovering the desired outcome states (e.g. HGG, CURROT, OUTPACE). It is summarized in Table 1 (column: Target dist. of curriculum). Therefore, we think it is a fair comparison since all algorithms do an undirected search by their own strategies before discovering the desired outcome states.

---

> > ### Comment · Area_Chair_KbKq · 2023-08-19
> > **please acknowledge author response**
> >
> > dear reviewer
> >
> > could you please let us know if you read the author response and if the response or the discussions here change your assessment?
> >
> > thanks
> >
> > AC

---

> > > ### Comment · Reviewer_WkJV · 2023-08-19
> > >
> > > Thanks a lot for your response. It confirms my scoring.

---

### Official Review · Reviewer_geTA · 2023-07-07

**Soundness:** 2 fair
**Presentation:** 2 fair
**Contribution:** 2 fair
**Rating:** 6
**Confidence:** 3

**Summary:**

This paper propose training multiple classifiers on a data set of "desired" and "reached but undesired" states, motivating these classifiers to be different off distribution, and using this divergence as a metric for exploration.

**Strengths:**

The approach is simple and easy to implement in many domains.

**Weaknesses:**

Since the approach is very similar to model-disagreement, I would expect a simple comparison to such an approach like exploration via disagreement.  Moreover, given that random-network distillation works so well for exploration, its unclear if the objective of the classifier here is really too important.  It may only be important that the metric has to do with the disagreement between two networks, and it may not matter what they are disagreeing about.

**Questions:**

What advantage does this approach have over generic model-disagreement, or random network distillation?

**Limitations:**

Given the number of existing approaches, and an unclear understanding of their relative limitations, it is unclear when this approach should be preferred over the others in a new domain.

---

> ### Author Rebuttal · Authors · 2023-08-08
>
> **Weaknesses:**
>
> 1. Since the approach is very similar to model-disagreement, I would expect a simple comparison to such an approach like exploration via disagreement. Moreover, given that random-network distillation works so well for exploration, its unclear if the objective of the classifier here is really too important. It may only be important that the metric has to do with the disagreement between two networks, and it may not matter what they are disagreeing about.
>
> → Thank you for your insightful comment. As you rightly pointed out, it is similar to model-disagreement in terms of encouraging the agent to explore uncertain areas. VDS, which utilizes the epistemic uncertainties of the value function ensembles to propose curriculum goal, is included as the baseline in the experiment section for the model-disagreement-based method. Also, Random Network Distillation (RND) encourages exploration by defining intrinsic reward based on the prediction error of the network rather than explicitly proposing the curriculum goal. However, VDS and RND do not have a convergence mechanism to the desired outcome states. That is, they just encourage endless exploration to find the novel states even after discovering the desired outcomes, which is not desirable for goal-reaching settings.
>
> Even though we can imagine the combination between external reward given from the environmental interaction and the RND's intrinsic reward, it still requires hyperparameter tuning for balancing the external (exploitation) and intrinsic reward (exploration). Also, since these methods do not consider multi-modal desired outcome distribution, they are prone to exploitation. That is, the agent does not explore other desired outcome states once it found a specific desired outcome distribution earlier. However, our proposed method keeps exploring when the agent is in the same setting because the curriculum cost for the undiscovered desired outcome state is not minimized yet. Furthermore, our method provides integrated methodology not only for the desired outcome-directed curriculum generation but also for the goal-conditioned intrinsic reward compared to VDS, and RND that require an assumption of access to the external reward.
>
> For the last concern, we would like to note that it matters what the classifiers are disagreeing. If the classifiers disagree on already explored states, efficient exploration toward the frontier is unavailable since the agent tries to reach the proposed curriculum goals in the already explored regions. Also, if the classifiers disagree on desired outcome states, the proposed curriculum goals will not be converged to the desired outcomes. It not only hinders the acceleration of the curriculum proposal toward the desired outcomes after discovering these but also prevents the agent from repeatedly practicing to reach the desired outcomes. Since we assume a setting where only some desired outcome examples are given before training begins and access to the desired outcome distribution and external reward are unavailable, convergence to the desired outcome states is required. Therefore, we think it matters that the classifiers disagree only on the unexplored states.
>
> We believe that these responses together adequately address this comment. However, if you still have remaining concerns or questions please let us know.
>
> **Questions:**
>
> 1. What advantage does this approach have over generic model-disagreement, or random network distillation?
>
> → Weakness section
>
> **Limitations:**
>
> 1. Given the number of existing approaches, and an unclear understanding of their relative limitations, it is unclear when this approach should be preferred over the others in a new domain.
>
> → Thank you for your comment. We added more details of the baseline as follows. (Also, there are conceptual comparisons between D2C and baselines in Table 1 and brief explanations are in lines 249-259. A quick review of these would be helpful to clarify the pros and cons of each different approach.)
>
> HGG and CURROT utilize Euclidean distance metric, they are prone to get stuck in an obstacle in an environment such as Maze, while our method D2C does not have such a problem.
>
> PLR and ALP-GMM leverage the regret or learning progress, which implicitly indicates the novelty or difficulty of the proposed curriculum. But these methods do not have a convergence mechanism to the desired outcome distribution, which induces endless exploration rather than achieving the given task. Furthermore, ALP-GMM depends on the Gaussian Mixture Model (GMM) that is susceptible to focusing on infeasible goals where the agent’s capability stagnates in the intermediate level of difficulty.
>
> OUTPACE is similar to our approach, but it requires Wasserstein distance-based temporal distance estimation for curriculum proposal, which can result in collapsed curriculum goals as described in lines 267-272. Also, for uncertainty quantification, it adopts meta-learning (MAML [1]) that requires gradient computation at every optimization iteration, while our method only requires a single neural network inference, leading to much faster curriculum optimization.
>
> Overall, the proposed method, D2C, has several advantages compared to other curriculum RL baselines, and we think it could be preferred over others in general.
>
> [1] Finn, Chelsea, Pieter Abbeel, and Sergey Levine. "Model-agnostic meta-learning for fast adaptation of deep networks." *International conference on machine learning*. PMLR, 2017.

---

> > ### Comment · Area_Chair_KbKq · 2023-08-19
> > **please acknowledge author response**
> >
> > dear reviewer
> >
> > could you please let us know if you read the author response and if the response or the discussions here change your assessment?
> >
> > thanks
> >
> > AC

---

> > ### Comment · Reviewer_geTA · 2023-08-21
> > **Response to Authors**
> >
> > I thank the authors for their detailed response.
> >
> > There are two critical arguments about the applicability of this approach which were brought up in the rebuttal but were not made clear in the original paper.  I think it is quite important for the paper to be reframed to make these points clear as these two points are make-or-break for a reader understanding the usefulness of the method.
> >
> > > However, VDS and RND do not have a convergence mechanism to the desired outcome states
> >
> > This is framing the "convergence to the desired outcome states" as a key criteria of what you would want out of a curriculum approach, which is indeed an important criteria the vast majority of other approaches overlook.  Making it it clear why this criteria is not just nice but absolutely necessary and making it a centerpiece of the work, embracing it as a key strength, would really strengthen the work.
> >
> > > Nevertheless, it is clear that addressing the “without external reward” framework itself is not only necessary but also increasingly crucial to effectively address RL problems that closely resemble real-world scenarios where access to the ground truth reward function is unavailable.
> >
> > This is framing the approach as not only tackling the curriculum problem but also the reward specification problem.  This should also be embraced as a strength.  If readers feel like it is just a "nice to have" criteria, then the domain-specific work from the designer seems hard to justify.  Ideally, you would show this by having a domain where you do not have a reward function and it is hard to specify one, but you can get the agent to perform the task anyway using your approach (doing a backflip in mujoco for instance, or other tasks in the DRL from human preferences space)
> >
> >
> > Without these re-framings, the choice made by the method to require the designer to provide data of desired states appear to be a severe limitation for a curriculum method.  While other curriculum methods are complete drop-in to a new environment, this one requires domain-specific work from the designer, which most RL researchers would rather avoid.
> >
> > With these framings, the choice made to use designer provided data is obviously necessary, and worth it because 1) providing a few examples is easier than providing a reward function, and 2) given that the designer has to specify something anyway, may as well specify these examples rather than a reward function because they also allow better curricula.
> >
> >
> > Thus it is very critical that the readers understand these two arguments from the paper. I suggest that these arguments be placed prominently in some form in the abstract/introduction so that they can not be easily overlooked.
> >
> > Given that what I had previously seen as weaknesses of this work I now see as strengths, I will be raising my score to a 6

---

> > > ### Author Response · Authors · 2023-08-21
> > >
> > > We sincerely appreciate your valuable insights and feedback. After conducting a thorough examination of our manuscript in light of the two pivotal points you highlighted, we found that these two arguments have been mentioned but their significance and implications seem to have been somewhat less emphasized. In line with your feedback, we are committed to revising the manuscript to better emphasize the core concept and benefits of the proposed method, thereby ensuring enhanced reader understanding. It would be really helpful to refine the quality of our work. Thank you for your constructive response.

---

### Official Review · Reviewer_pHjM · 2023-07-11

**Soundness:** 3 good
**Presentation:** 3 good
**Contribution:** 3 good
**Rating:** 6
**Confidence:** 3

**Summary:**

Focusing on goal-conditioned RL, the author propose Diversify for Disagreement & Conquer (D2C) to form outcome-directed exploration by generating a sequence of curriculum goals, given some desired outcome examples. By ensuring multiple classifiers disagree on unseen states, D2C uses bipartite matching to create curriculum goals, which are interpolated between the initial state distribution and arbitrarily distributed desired outcome states for enabling the agent to conquer the unexplored region. The experiments demonstrate that D2C surpasses previous curriculum RL methods in goal-conditioned RL experiments.

**Strengths:**

- I appreciate the idea to use the diversifying functions for disagreement on underspecified data and to quantify the similarity between the visited states and desired outcome states. To use the pseudo probability as the intrinsic reward is also straightforward. The proposed method is simple but effective.

- The paper is well written.


**Weaknesses:**

- It is not clear for me how to interpolate the curriculum distribution from the initial state distribution through minimizing Eq 4.
- Some related work and its comparison missing. Diversity has been considered in Curriculum RL. For example, Curriculum-guided Hindsight Experience Replay (NeurIPS 2019).
- It is not open sourced.

**Questions:**

- Please clarify the interpolation process as referenced in weakness.

- Figure 6 can be better if the labels do not cover the curves. The font sizes in the figures are too small to read.

---

> ### Author Rebuttal · Authors · 2023-08-08
>
> **Weaknesses:**
> 1. It is not clear for me how to interpolate the curriculum distribution from the initial state distribution through minimizing Eq 4.
>
> → Question section.
>
> 2. Some related work and its comparison missing. Diversity has been considered in Curriculum RL. For example, Curriculum-guided Hindsight Experience Replay (NeurIPS 2019).
>
> → Thank you for your suggestion about the related work. We had a quick check of the CHER paper, and there exist some related keywords, so we will add this work in the revised version. But, in terms of methodology, there are several key differences between CHER and our method.
>
> First, CHER is about goal relabeling technique like HER during the RL update phase rather than proposing curriculum goal during the rollout phase. That is, CHER is appropriate for an add-on-module on our method rather than a direct comparison since our method is also based on the HER as described in Algorithm 1 in Appendix B.
>
> Second, as far as we understand CHER, the distance metric in the proximity measurement is based on the Euclidean distance metric, which is not appropriate for the geometry-agnostic curriculum proposal property. Also, diversity in CHER means how representative sampled curriculum candidates are with respect to the states in the replay buffer. However, diversity in D2C induces disagreement between the classifiers for the unlabeled target data, and it is reflected in curriculum optimization cost to encourage the algorithm to propose curriculum goals into the unexplored region. Thus, the same keyword “diversity” is somewhat different between our work and CHER.
>
> Due to the short period of the rebuttal phase, we do not have enough time to experiment with CHER. But, based on the results in the CHER paper, we think that CHER has contributed to performance increases compared to RL+HER rather than making the algorithm work in an environment where RL+HER completely fails. Since most of the tasks used in D2C cannot be solved with RL+HER without additional tools such as curriculum proposal (based on our own experimental results), we expect that CHER will not bring significant improvement compared to RL+HER in the tasks used in D2C.
>
> 3. It is not open sourced.
>
> → The source code is already included in the supplementary material. Also, the code will be open-sourced after the decision.
>
> **Questions:**
> 1. Please clarify the interpolation process as referenced in weakness.
>
> →Thank you for your comment. The pseudo probability in Eq (3) represents how similar the queried state is to the explored states or desired outcome states (lines 178-188), and disagreement between classification heads is represented as intermediate values between the 0 (explored states) and 1 (desired outcome states) when optimizing the curriculum loss in Eq (4).
> For example, before discovering the desired outcome states, the batch data sampled from the replay buffer in Eq (4) include a diverse range of states from frequently visited states to rarely visited ones such as the states in the frontier of the explored region. As described in lines 178-188, $P_{pseudo}(y=1|g^+)$ is nearly 1 and the loss will be large for the frequently visited states as all classifiers correctly classify them as a label 0 (i.e. $P_{pseudo}(y=1|s)=0$), and the loss will be small for the states in the frontier since the classifiers disagree due to the diversification (i.e. $P_{pseudo}(y=1|s)$ is larger than 0). Thus, the curriculum objective encourages selecting the states in the frontier region.
> After discovering the desired outcome states, the curriculum objective encourages selecting the states close to the desired outcome state since the corresponding loss will be the smallest among the losses of states in the replay buffer.
>
> In the case of multi-modal desired outcome distribution as described in Fig 2 (two desired outcome examples, K=2 in Eq (6)), the conditional classifier enables non-collapsed curriculum proposals even when the agent achieves a specific desired outcome earlier, since we utilize bipartite matching (i.e. no duplicative selection) to find the curriculum goal candidates in the replay buffer.
>
> For clarity, we would like to note that cross-entropy in Eq (4) is used as a convex function that decreases as $P_{pseudo}(y=1|s)$ increases, not as an exact entropy minimization to match the probability distribution (because $P_{pseudo}$ in Eq (3) is not mathematically probability, so we coined it with “pseudo”).
>
> 2. Figure 6 can be better if the labels do not cover the curves. The font sizes in the figures are too small to read.
>
> → Thank you for your suggestion. We modified the location of the labels and font size of Figure 6, and we attached the pdf with modified figures of experimental results at the global response. Due to the single-page limit, current figures are a little bit small. After the decision, we will further modify the figures with better visibility and readability on the additional page.

---

### Official Review · Reviewer_LCMf · 2023-07-25

**Soundness:** 2 fair
**Presentation:** 2 fair
**Contribution:** 2 fair
**Rating:** 4
**Confidence:** 4

**Summary:**

The paper proposed Diversify for Disagreement & Conquer (D2C), which is an outcome-directed curriculum RL method. D2C performs diversification of the goal-conditional classifiers to identify similarities between visited and desired outcome states and ensures that the classifiers disagree on states from out-of-distribution, which enables quantifying the unexplored region and designing an arbitrary goal-conditioned intrinsic reward signal in a simple and intuitive way. D2C then employs bipartite matching to define a curriculum learning objective that produces a sequence of well-adjusted intermediate goals, which enable the agent to automatically explore and conquer the unexplored region.

**Strengths:**

D2C performs a classifier diversification process to distinguish the unexplored region from the explored area and desired outcome example.

D2C conquers the unexplored region by proposing the curriculum goal and the shaped intrinsic reward.

D2C enables the agent to automatically progress toward the desired outcome states without prior knowledge of the environment.

**Weaknesses:**

1. The writing of the paper needs some polishing to improve its readability.

2. The settings of the experiments, e.g. the map types and sizes, are not challenging to show the significance of the proposed model.

3. The complexity analysis is missing. Empirically, it takes 1 - 2 days to train a solution for a 36 * 36 map as described in Appendix.

**Questions:**

1. OUTPACE is the latest model in comparison, but it performs (near) the worst in all tasks (Fig.4 & 5). I had a quick check of OUTPACE paper, and it reports completely different results. Please explain how this could happen.

2. What is the complexity of the proposed model?

**Limitations:**

The paper has a good summary of its limitations and potential societal impact.

---

> ### Author Rebuttal · Authors · 2023-08-08
>
> **Weakness:**
> 1. The writing of the paper needs some polishing to improve its readability.
>
> → Thank you for your suggestion. Since there is no way to revise the current manuscript in the rebuttal phase, we will try to rewrite some phrases and fix some grammatical errors as much as possible after the decision. If you specify some examples, it would be really helpful for us to revise our work with better readability.
>
> 2. The settings of the experiments, e.g. the map types and sizes, are not challenging to show the significance of the proposed model.
>
> → Thank you for your comment. We would like to note that we followed the settings of the experiments in prior curriculum RL works. We think the environments used in this work are already challenging enough to show the significance of our model since it outperforms other prior state-of-the-art works in these environments. In other words, only D2C can achieve all the tasks regardless of their domain, geometry, distribution of the desired outcome states, etc. If you propose a specific environment that is appropriate to show the significance of our model, we will try to experiment and evaluate our method. It would be really helpful to improve our work’s contribution.
>
> 3. The complexity analysis is missing. Empirically, it takes 1 - 2 days to train a solution for a 36 * 36 map as described in Appendix.
>
> → Question section
>
>
> **Questions:**
> 1. OUTPACE is the latest model in comparison, but it performs (near) the worst in all tasks (Fig.4 & 5). I had a quick check of OUTPACE paper, and it reports completely different results. Please explain how this could happen.
>
> → Thank you for your comment about the baseline. As described in lines 267-272, OUTPACE uses Wasserstein distance-based temporal distance estimation to propose curriculum goals into the temporally distant region from the initial state distribution. Once the agent starts to explore a temporally far region in a specific direction earlier, the curriculum proposal is prone to be collapsed toward this area (Figure 3). Thus, there exists only one way to the desired outcome state (i.e. uni-modal desired outcome distribution) in the environments used in OUTPACE to prevent such a problem. Also, the curriculum objective in OUTPACE includes meta-learning-based uncertainty quantification to combine with the temporal distance estimation, which results in extensive hyperparameter tuning for numerical stability and balancing the trade-off between uncertainty and temporal distance. In practice, without such tuning, we found that OUTPACE’s loss frequently blows up. On the contrary, our method simply consists of a single module (i.e. classifiers for disagreement) with cross-entropy loss and mutual information computation, which results in a stable training process compared to the meta-learning process and Wasserstein distance estimation in OUTPACE.
>
> 2. What is the complexity of the proposed model?
>
> → Thank you for your comment about the complexity analysis. Since our method is based on simple MLP-based classifiers with multiple heads, the inference time of the classifier is negligible. The classifier training is performed every 2000~4500 steps as described in Table 3 in Appendix A. Therefore, classifier training time is nearly negligible compared to the SAC’s update which is performed at every step. The curriculum optimization itself requires solving a bipartite matching problem as described in section 4.3, and this is addressed via the Minimum Cost Maximum Flow algorithm, which is implemented as a few lines of c code adopted from prior work [1]. It consists of a double for-loop. The first loop is iterations on a few trajectories sampled from the replay buffer, and the second loop is iterations on the desired goals. In practice, this takes around 10 seconds, but curriculum optimization is also performed every thousands of steps (it depends on the maximum episode horizon of each environment), thus, the overall curriculum optimization time is not a significant bottleneck. We believe that these responses adequately address this comment about the complexity. However, if you still have remaining concerns or questions please let us know.
>
> [1] Ren, Zhizhou, et al. "Exploration via hindsight goal generation." *Advances in Neural Information Processing Systems* 32 (2019).

---

> > ### Comment · Reviewer_LCMf · 2023-08-18
> >
> > I thank the authors for their detailed response. I appreciate their efforts and their response did help demystify some aspects of the paper. But I still honestly feel that the paper needs some important improvement. So I'll keep my current overall recommendation.

---

> > > ### Author Response · Authors · 2023-08-19
> > >
> > > Thank you for your response. We appreciate you engaging in the discussion.
> > >
> > > We believe we have made sincere efforts to address the issues and clarified some concepts that may have been misunderstood by the reviewers due to our previous unclear description. We politely request the reviewers to provide more clear and specific suggestions for important improvement if possible. We sincerely appreciate their valuable assistance in improving the quality of our work.

---

### Author Rebuttal · Authors · 2023-08-08

We sincerely thank all the reviewers for reviewing our work and providing constructive feedback. We hope that our response has adequately addressed your comments. If you have any remaining questions (existing or new ones) that we can address in our follow-up response to improve your opinion about our work, please do not hesitate to provide additional feedback in the comments. It would be greatly appreciated if we could have more discussions about our work which would provide valuable insights towards further developing our research into a meaningful contribution in the RL domain.

Also, there is an ask for figures with large font sizes, so we attached modified figures in the pdf file.

---

### Decision · Program_Chairs · 2023-09-21

**Decision:**

Accept (poster)

**Comment:**

The paper studies an outcome-directed curriculum RL method, which encourages disagreement on underspecified data in order to improve the explore vs exploit trade-off in large state spaces. This diversity mechanism enables quantifying (and rewarding exploration of) the unexplored region, by ensuring that classifiers disagree on states from out-of-distribution. The method then proposes a curriculum learning objective that produces a sequence of intermediate goals, which enable the agent to explore and conquer unexplored regions.

Reviewer consensus after the discussion phase is that this is an interesting contribution. We hope the detailed feedback can help to clarify the camera-ready version of the paper.